# A D-2-hydroxyglutarate biosensor based on specific transcriptional regulator DhdR

Dan Xiao[1,5], Wen Zhang [2,5], Xiaoting Guo[3], Yidong Liu[1], Chunxia Hu[1], Shiting Guo[1], Zhaoqi Kang[1], Xianzhi Xu[1], Cuiqing Ma[1], Chao Gao [1✉] & Ping Xu [4✉]

D-2-Hydroxyglutarate (D-2-HG) is a metabolite involved in many physiological metabolic processes. When D-2-HG is aberrantly accumulated due to mutations in isocitrate dehydrogenase or D-2-HG dehydrogenase, it functions in a pro-oncogenic manner and is thus considered a therapeutic target and biomarker in many cancers. In this study, DhdR from *Achromobacter denitrificans* NBRC 15125 is identified as an allosteric transcriptional factor that negatively regulates D-2-HG dehydrogenase expression and responds to the presence of D-2-HG. Based on the allosteric effect of DhdR, a D-2-HG biosensor is developed by combining DhdR with amplified luminescent proximity homogeneous assay (AlphaScreen) technology. The biosensor is able to detect D-2-HG in serum, urine, and cell culture medium with high specificity and sensitivity. Additionally, this biosensor is used to identify the role of D-2-HG metabolism in lipopolysaccharide biosynthesis of *Pseudomonas aeruginosa*, demonstrating its broad usages.

[1] State Key Laboratory of Microbial Technology, Shandong University, Qingdao, People's Republic of China. [2] Institute of Medical Sciences, The Second Hospital, Cheeloo College of Medicine, Shandong University, Jinan, People's Republic of China. [3] Eye Hospital of Shandong First Medical University, Jinan, People's Republic of China. [4] State Key Laboratory of Microbial Metabolism, Joint International Research Laboratory of Metabolic & Developmental Sciences, and School of Life Sciences & Biotechnology, Shanghai Jiao Tong University, Shanghai, People's Republic of China. [5] These authors contributed equally: Dan Xiao, Wen Zhang. ✉email: jieerbu@sdu.edu.cn; pingxu@sjtu.edu.cn

D -2-Hydroxyglutarate (D-2-HG) has traditionally been considered an abnormal metabolite associated with the neurometabolic disorder D-2-hydroxyglutaric aciduria (D-2-HGA)[1]. However, recent studies have revealed the accumulation of D-2-HG in many tumor cells due to mutations in isocitrate dehydrogenase 1/2 (IDH1/2)[2–6]. Thus, D-2-HG is also considered an oncometabolite. It has been reported that D-2-HG may also be an important metabolic intermediate involved in different core metabolic processes, including L-serine synthesis[7], lysine degradation[8], and 4-hydroxybutyrate catabolism[9]. D-2-HG dehydrogenase (D2HGDH) catalyzes the conversion of D-2-HG to 2-ketoglutarate (2-KG) and is the key enzyme involved in D-2-HG catabolism[7]. D-2-HG is present at rather low levels (< 0.9 μM in the plasma of healthy humans) under physiological conditions[1,10]. This suggests that organisms may have evolved specific mechanisms to respond to D-2-HG accumulation by increasing D-2-HG catabolism through enhanced D2HGDH expression[11–14]. However, the regulation of D2HGDH expression is not fully understood, and the mechanism whereby organisms responding to the presence of D-2-HG has not yet been elucidated.

D-2-HG levels are increased in patients with D-2-hydroxyglutaric aciduria and *IDH* mutation-related cancers[1,2]. As such, the detection of D-2-HG is relevant for the diagnosis and monitoring of these diseases[15–17]. Gas chromatography-tandem mass spectrometry (GC-MS/MS)[18,19] and liquid chromatography-tandem mass spectrometry (LC-MS/MS)[20,21] are often used to quantitatively assess D-2-HG levels. Chiral derivatization with proper reagents is necessary to distinguish D-2-HG from its mirror-image enantiomer L-2-HG by GC-MS/MS and LC-MS/MS[22]. Ion mobility-quadrupole-time-of-flight mass spectrometry (IM-QTOF-MS), a powerful tool to separate complex mixtures, may resolve both enantiomers without any derivatization[23]. On the other hand, allosteric transcription factors (aTFs) in bacteria have evolved to sense a variety of chemicals[24]. Various aTFs, such as OtrR, HosA, HucR, and SRTF1, have been well characterized, and then are coupled with various transduction systems to develop convenient biosensors for detection of oxytetracycline, 4-hydroxybenzoic acid, uric acid, and progesterone[25–28]. To date, no aTF that specifically responds to D-2-HG has been identified, restricting the development of D-2-HG biosensors.

In this study, we identify a D-2-HG catabolism regulator DhdR in *Achromobacter denitrificans* NBRC 15125. This aTF can repress expression of the D2HGDH-encoding gene *d2hgdh*, and is specifically derepressed by D-2-HG. Then, we combine DhdR with amplified luminescent proximity homogeneous assay (AlphaScreen) technology, a bead-based immunoassay, to develop a D-2-HG biosensor with high specificity and sensitivity. We select various biological samples to demonstrate applicability of the biosensor. In addition, we use the biosensor to identify UDP-2-acetamido-2-deoxy-D-glucuronic acid (UDP-GlcNAcA) 3-dehydrogenase (WbpB) of *Pseudomonas aeruginosa* PAO1 as a D-2-HG anabolic enzyme that also participates in the intracellular generation of D-2-HG from 2-KG.

## Results

**Genome analysis predicts a D-2-HG catabolism operon.** Bacteria have evolved various aTFs to respond to different stimuli. To identify aTFs that respond to D-2-HG and regulate the expression of D2HGDH, two open reading frames (ORFs) of upstream and two ORFs of downstream of *d2hgdh* in bacteria containing *d2hgdh* were subjected to the gene occurrence profile analysis. Genes encoding GntR family transcriptional regulators, electron transfer flavoprotein (ETF), 3-phosphoglycerate dehydrogenase (SerA), carbohydrate diacid regulator (CdaR), lactate permease

(LldP), and glycolate oxidase iron-sulfur subunit (GlcF) appear to be the most frequently observed in the neighborhood of *d2hgdh* (Supplementary Table 1).

Five typical patterns of organized chromosomal clusters were identified through subsequent chromosomal gene clustering analysis (Fig. 1a). The genes *gntR* and *cdaR* are located upstream of *d2hgdh* in *Parageobacillus thermoglucosidasius* DSM 2542 and *Bacillus cereus* NJ-W, respectively. However, the genes *lldP* and *glcF*, which respond to the metabolism of two other hydroxycarboxylic acids, lactate and glycolate, are also located adjacent to *d2hgdh* in *P. thermoglucosidasius* DSM 2542 and *B. cereus* NJ-W. Thus, these two transcriptional regulators in *P. thermoglucosidasius* DSM 2542 and *B. cereus* NJ-W may respond to D-2-HG and regulate D2HGDH expression but lactate and glycolate may also be their effectors. On the other hand, *gntR* is located directly upstream of *d2hgdh* in *A. denitrificans* NBRC 15125, and no adjacent hydroxycarboxylic acid metabolism-related protein is encoded. Thus, we hypothesized that the genes *gntR* and *d2hgdh* may comprise an operon that is specifically responsible for D-2-HG catabolism in *A. denitrificans* NBRC 15125. The uncharacterized transcriptional regulator GntR was tentatively designated as D-2-hydroxyglutarate dehydrogenase regulator (DhdR).

**D2HGDH is required for extracellular D-2-HG metabolism.** D2HGDH of *A. denitrificans* NBRC 15125 was expressed as a His$_6$-tagged protein in *Escherichia coli* BL21(DE3) and purified by affinity chromatography. Sodium dodecyl sulfate-polyacrylamide gel electrophoresis (SDS-PAGE) and size exclusion chromatography revealed that D2HGDH is a dimer (Fig. 1b and Supplementary Fig. 1). Substrate screening revealed that D2HGDH had high substrate specificity and only exhibited distinct activity towards D-2-HG (Fig. 1c). The product of D2HGDH-catalyzed dehydrogenation of D-2-HG was identified as 2-KG by high-performance liquid chromatography (HPLC) analysis (Fig. 1d). The apparent $K_m$, $V_{max}$, and $k_{cat}$ of purified D2HGDH for D-2-HG were $31.16 \pm 1.41$ μM, $40{,}682 \pm 946$ μM min$^{-1}$, and $6.90 \pm 0.16$ s$^{-1}$, respectively (Supplementary Fig. 2 and Supplementary Table 2).

SerA, which can catalyze the conversion of 2-KG to D-2-HG, is the key enzyme for L-serine biosynthesis in various species, such as *P. stutzeri* A1501, *P. aeruginosa*[7], *E. coli*[29], *Saccharomyces cerevisiae*[30], and *Homo sapiens*[31]. In our study, SerA of *A. denitrificans* NBRC 15125 was overexpressed, purified, and identified to be a tetramer (Supplementary Fig. 3a, b). It was also able to reduce 2-KG to D-2-HG (Supplementary Fig. 3c, d), and the apparent $K_m$, $V_{max}$, and $k_{cat}$ for 2-KG were $57.88 \pm 0.71$ μM, $4{,}062 \pm 42$ μM min$^{-1}$, and $5.79 \pm 0.06$ s$^{-1}$, respectively (Supplementary Fig. 3e and Supplementary Table 2). However, *A. denitrificans* NBRC 15125 (Δ*d2hgdh*), *A. denitrificans* NBRC 15125 (Δ*serA*), and *A. denitrificans* NBRC 15125 (Δ*d2hgdh*Δ*serA*) did not exhibit either growth defects in Luria-Bertani (LB) medium or accumulation of extracellular D-2-HG (Fig. 1e and Supplementary Fig. 4). No significant differences in intracellular D-2-HG concentrations were found between *A. denitrificans* NBRC 15125 and its derivatives (Fig. 1e). These results demonstrate that there is no SerA-catalyzed endogenous D-2-HG production in *A. denitrificans* NBRC 15125, and that D2HGDH does not participate in endogenous D-2-HG catabolism. *A. denitrificans* NBRC 15125 (Δ*serA*) and *A. denitrificans* NBRC 15125 (Δ*d2hgdh*Δ*serA*) were able to grow well on solid minimal medium without exogenous L-serine (Fig. 1f). It should be noted that *A. denitrificans* NBRC 15125 is able to use D-2-HG as the sole carbon source for growth. Disruption of *d2hgdh* abolished the ability of *A. denitrificans* NBRC 15125 (Δ*d2hgdh*) to assimilate D-2-HG but did not affect growth in the presence of L-2-HG, D-malate, D-lactate, and 2-KG, indicating

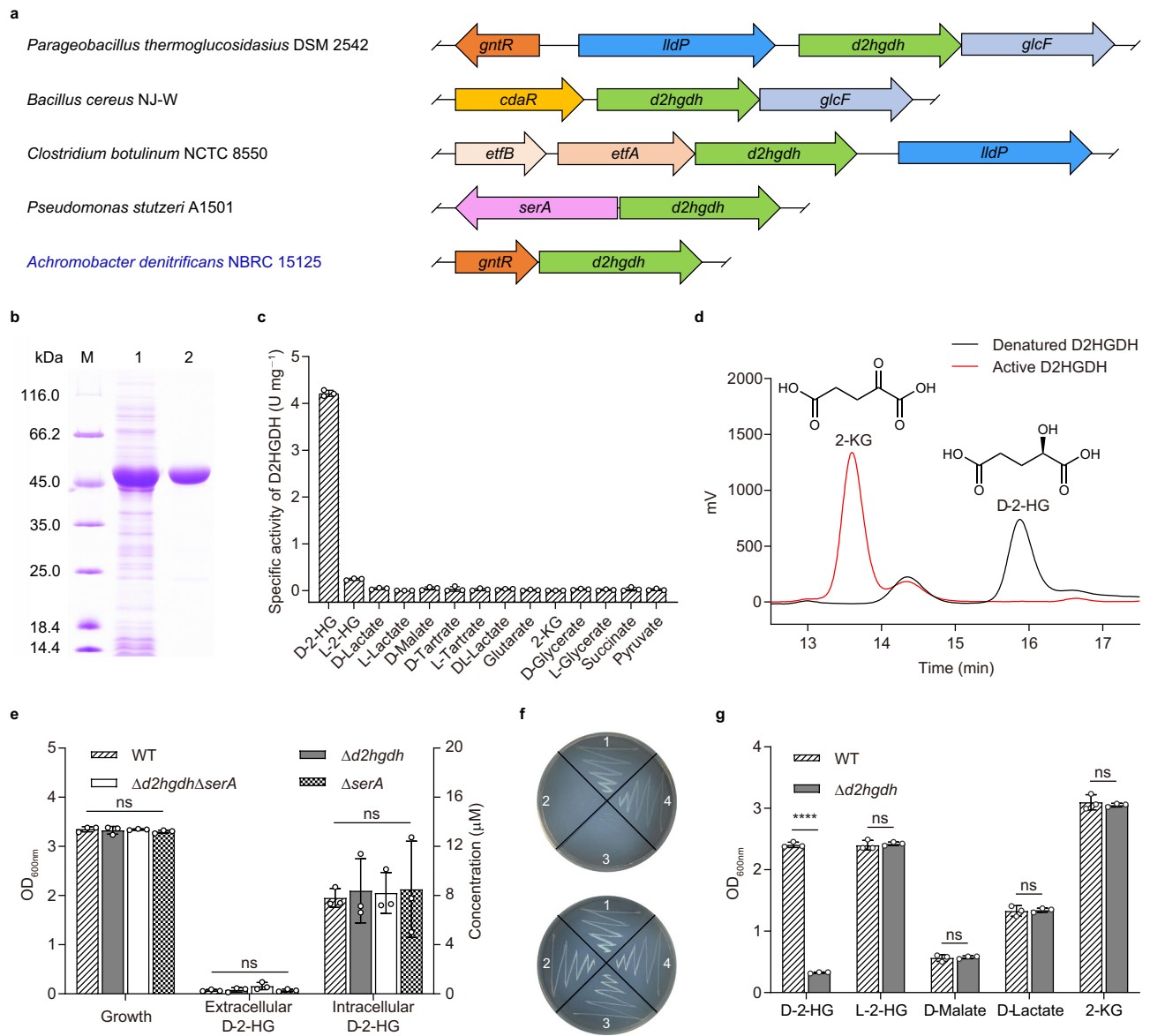

**Fig. 1 D2HGDH contributes to D-2-HG catabolism in *A. denitrificans* NBRC 15125. a** Schematic representation of genome context analysis of *d2hgdh* in different species. Orthologs are shown in the same color. Arrows indicate the direction of gene translation. The corresponding proteins encoded by the indicated genes are D-2-HG dehydrogenase (*d2hgdh*), GntR family regulator protein (*gntR*), lactate permease (*lldP*), glycolate oxidase iron-sulfur subunit (*glcF*), carbohydrate diacid regulator (*cdaR*), electron transfer flavoprotein alpha/beta-subunit (*etfA/B*), and 3-phosphoglycerate dehydrogenase (*serA*). **b** SDS-PAGE analysis of purified D2HGDH in *A. denitrificans* NBRC 15125. Lane M, molecular weight markers; lane 1, crude extract of *E. coli* BL21(DE3) harboring pETDuet-*d2hgdh*; lane 2, purified D2HGDH using a HisTrap column. **c** Substrate specificity of D2HGDH in *A. denitrificans* NBRC 15125. Data shown are mean ± standard deviations (s.d.) (*n* = 3 independent experiments). **d** HPLC analysis of the product of D2HGDH-catalyzed D-2-HG dehydrogenation. The reaction mixtures containing D-2-HG (1 mM), 3-(4,5-dimethylthiazol-2-yl)-2,5-diphenyltetrazolium bromide (MTT, 5 mM), and active or denatured D2HGDH (1 mg mL$^{-1}$) in 50 mM Tris-HCl (pH 7.4) were incubated at 37 °C for 30 min. Black line, the reaction with denatured D2HGDH; red line, the reaction with active D2HGDH. **e** Growth, extracellular, and intracellular D-2-HG concentrations of *A. denitrificans* NBRC 15125 and its derivatives cultured in LB medium at mid-log stage. Data shown are mean ± s.d. (*n* = 3 independent experiments). **f** Growth of *A. denitrificans* NBRC 15125 and its derivatives on solid minimal medium containing 5 g L$^{-1}$ D-2-HG (top) or 2-KG (bottom) as the sole carbon source after 36 h. 1. *A. denitrificans* NBRC 15125; 2. *A. denitrificans* NBRC 15125 (Δ*d2hgdh*); 3. *A. denitrificans* NBRC 15125 (Δ*d2hgdh*Δ*serA*); 4. *A. denitrificans* NBRC 15125 (Δ*serA*). **g** Growth of *A. denitrificans* NBRC 15125 and *A. denitrificans* NBRC 15125 (Δ*d2hgdh*) in minimal medium containing different carbon sources after 24 h. Data shown are mean ± s.d. (*n* = 3 independent experiments). The significance was analyzed by a two-tailed, unpaired *t*-test. ****$P$ < 0.0001; ns, no significant difference ($P \geq 0.05$). Exact $P$ values are reported in the Source Data file. Source data are provided as a Source Data file.

that D2HGDH is critical for the utilization of exogenous D-2-HG by this strain (Fig. 1g).

**DhdR represses *d2hgdh* expression and responds to D-2-HG.** The *dhdR* and *d2hgdh* genes are adjacent to each other in the

genome of *A. denitrificans* NBRC 15125. To determine whether *dhdR* and *d2hgdh* form an operon, a 301-bp fragment overlapped with *dhdR* and *d2hgdh* was amplified by reverse transcription-PCR (RT-PCR) using cDNA generated from the total RNA of cells grown in minimal medium containing 3 g L$^{-1}$ D-2-HG or pyruvate as the sole carbon source. As shown in Fig. 2a, the

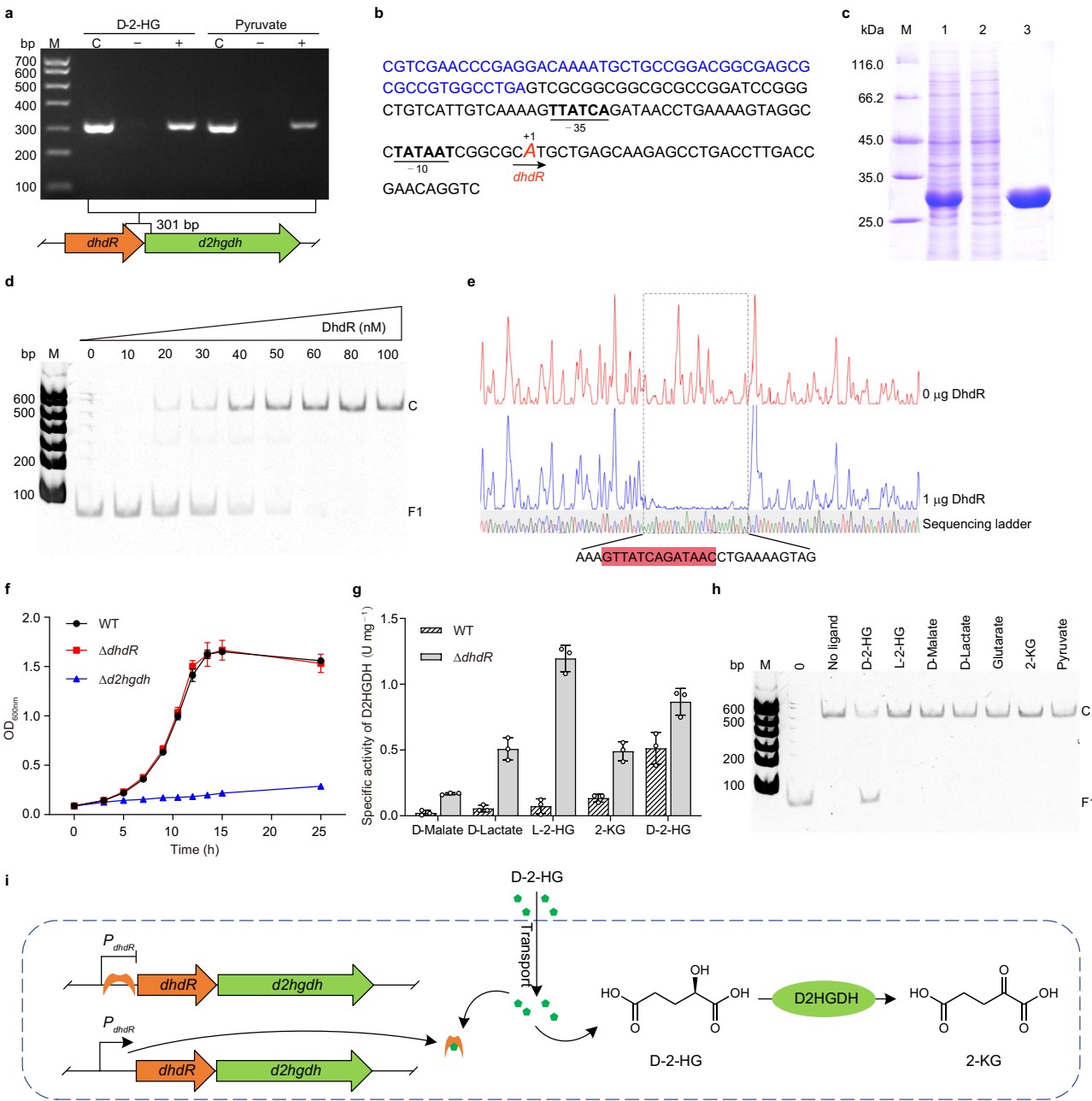

**Fig. 2 DhdR negatively regulates the catabolism of D-2-HG. a** Identification of the cotranscription of *dhdR* and *d2hgdh* in the presence of D-2-HG or pyruvate by RT-PCR. Lanes C, genomic DNA as template; lanes +, cDNA as template; lanes −, RNA as template; lane M, DNA ladder marker. The amplicon locations are indicated in panel **a**. **b** Map of the intergenic region upstream of *dhdR-d2hgdh* operon. Transcriptional start site (TSS) is shown in enlarged red letter and marked by +1. The putative −35 and −10 regions are shown in bold and underlined. The translational start codon of *dhdR* is shown in italics. **c** SDS-PAGE analysis of purified DhdR. Lane M, molecular weight markers; lane 1, crude extract of *E. coli* BL21(DE3) harboring pETDuet-*dhdR*; lane 2, the unbound protein of the HisTrap HP column; lane 3, purified DhdR using a HisTrap column. **d** EMSAs with the 81-bp fragment (F1) upstream of *dhdR* (10 nM) and purified DhdR (0, 10, 20, 30, 40, 50, 60, 80, and 100 nM). C, complex of DhdR-F1. **e** DNase I footprinting analysis of DhdR binding to the *dhdR* promoter region. Sequences of DhdR protected region are shown at the bottom and the palindrome sequences are indicated with a red box. **f** Growth of *A. denitrificans* NBRC 15125 and its derivatives cultured in minimal medium containing 3 g L⁻¹ D-2-HG. Data shown are mean ± s.d. (*n* = 3 independent experiments). **g** Activity of D2HGDH in *A. denitrificans* NBRC 15125 and *A. denitrificans* NBRC 15125 (Δ*dhdR*) cultured in minimal medium with different compounds as the sole carbon source. Data shown are mean ± s.d. (*n* = 3 independent experiments). **h** D-2-HG prevents DhdR binding to F1. The leftmost lane without DhdR (0) shows the migration of free DNA. Lane M, DNA ladder marker. C, complex of DhdR-F1. **i** Proposed model for the regulation of D-2-HG catabolism by DhdR in *A. denitrificans* NBRC 15125. DhdR represses the expression of *dhdR-d2hgdh* genes. D-2-HG is the effector of DhdR and prevents DhdR binding to the *dhdR* promoter region. Source data are provided as a Source Data file.

intergenic *dhdR-d2hgdh* were successfully amplified, suggesting that these two genes are co-transcribed. The transcriptional start site (TSS) of *dhdR* was identified by rapid amplification of cDNA ends (RACE). The TSS was confirmed to be an adenine (A) residue that overlaps with the *dhdR* start codon, and the putative −10 (TATAAT) and −35 (TTATCA) regions are separated by 20 bp (Fig. 2b and Supplementary Fig. 5).

The His$_6$-tagged DhdR was expressed and purified by affinity chromatography (Fig. 2c). Size exclusion chromatography and SDS-PAGE indicate that DhdR behaves as a dimer (Supplementary Fig. 6). An electrophoretic mobility shift assay (EMSA) was performed using purified DhdR and the 81-bp fragment (F1) upstream of *dhdR*. As shown in Fig. 2d, DhdR formed a complex with F1 in a concentration-dependent manner and a complete gel shift was observed with 6-fold molar excess of DhdR. The formation of DhdR-F1 complexes was proved to be specific since the non-specific DNA fragment, an internal fragment of *dhdR*, could not interact with DhdR (Supplementary Fig. 7). A DNase I footprinting assay was also performed using purified DhdR and F1 end-labeled with 6-carboxyfluorescein (FAM), and a clearly protected region containing a motif with a palindromic sequence (5′-GTTATCAGATAAC-3′) was identified (Fig. 2e). The protected region overlaps with the putative −35 region relative to the TSS of *dhdR*, implying that DhdR functions as a repressor.

As shown in Fig. 2f, *A. denitrificans* NBRC 15125 (Δ*dhdR*) was able to grow in the presence of D-2-HG, as was the wild-type strain, whereas *A. denitrificans* NBRC 15125 (Δ*d2hgdh*) was not. Additionally, the D2HGDH activity of *A. denitrificans* NBRC 15125 (Δ*dhdR*) cultured in medium containing D-malate, D-lactate, L-2-HG, or 2-KG was much higher than that of *A. denitrificans* NBRC 15125, further supporting that DhdR represses D2HGDH expression (Fig. 2g).

D2HGDH activity in *A. denitrificans* NBRC 15125 was significantly increased when cultured in medium containing D-2-HG (Fig. 2g), suggesting that D-2-HG can induce D2HGDH expression. The effects of L-2-HG, D-malate, D-lactate, glutarate, 2-KG, pyruvate, and D-2-HG on the binding of DhdR to F1 were also assessed by EMSA. As shown in Fig. 2h, only D-2-HG could inhibit the binding of DhdR to F1. In addition, a fluorescence-based thermal shift (FTS) assay was performed to identify the ligand that binds to DhdR. Among the 30 compounds (100 μM) tested, only D-2-HG could lead to a significant increase of Tm relative to that of DhdR without ligands (ΔTm = 7.50 °C) (Supplementary Fig. 8). The result of isothermal titration calorimetry (ITC) also indicates that D-2-HG binds to DhdR ($K_D = 1.16 \pm 0.16$ μM) (Supplementary Fig. 9). Therefore, we propose a model that DhdR negatively regulates the D-2-HG catabolism, and D-2-HG is the specific effector of DhdR in *A. denitrificans* NBRC 15125 (Fig. 2i).

**DhdR behaves as a sensing element to develop biosensors**. The specific response of DhdR to D-2-HG inspired us to develop a D-2-HG biosensor with DhdR as the sensing element. The AlphaScreen technology was selected as the transducing element of this biosensor. As shown in Fig. 3a, a biotinylated DNA fragment [Bio-*dhdO* that contains the DhdR protected region (5′-GTTATCAGATAAC-3′)], and His$_6$-tagged DhdR are bound to streptavidin donor and nickel chelate acceptor beads, respectively. D-2-HG can prevent the binding of DhdR to Bio-*dhdO* and increase the distance from donor beads to acceptor beads. Therefore, the singlet oxygen ($^1O_2$) generated from the conversion of ambient oxygen upon laser excitation of donor beads at 680 nm cannot be transferred to acceptor beads, resulting in reduced luminescence signals at 520–620 nm. Through this approach, the ambient concentration of D-2-HG can be quantified

through the measurable luminescence signals. A formal D-2-HG measurement process includes diluting the biological samples (if necessary), incubating Bio-*dhdO* and His$_6$-tagged DhdR with commercial streptavidin donor and nickel chelate acceptor beads, and detecting luminescence signals with a laboratory plate reader (Supplementary Fig. 10).

To demonstrate the feasibility of this approach, the oscillations of the relative luminescence unit (RLU) induced by the Bio-*dhdO*-DhdR interaction were assayed under a variety of conditions. As shown in Fig. 3b, almost no luminescence signal was observed in the reaction system with only 1 nM Bio-*dhdO*. Upon the addition of DhdR, the luminescence signal appeared and increased as the concentration of DhdR increased (0.01–1 nM). The addition of D-2-HG to induce the dissociation of Bio-*dhdO* and DhdR led to a significantly decreased luminescence intensity, as expected. Subsequently, the reaction system was optimized for the quantification of D-2-HG. HBS-P buffer, which gives the highest signal-to-noise ratio (S/N) and lowest background signal (Supplementary Fig. 11), was selected to be the optimal working buffer for D-2-HG quantification. Additionally, this buffer has a high buffering capacity across a range of sample pH values (pH 4.0–10.0) (Supplementary Fig. 12). Various inorganic substances or amino acids (100 μM) were determined to not interfere with the RLU of the biosensor (Supplementary Table 3). Next, the concentrations of DhdR and Bio-*dhdO* were optimized through cross-titration. As shown in Fig. 3c, a hook effect appeared when the concentration of Bio-*dhdO* was higher than 1 nM. A credible signal ($3.70 \times 10^5$ RLU, S/N = 151.24) could be acquired when the concentrations of DhdR and Bio-*dhdO* were 0.3 and 1 nM, respectively. Thus, 0.3 nM of DhdR and 1 nM of Bio-*dhdO* were used in the original D-2-HG biosensor B$_{D2HG}$-0, and the luminescence signals emitted at a series of D-2-HG concentrations were measured. As shown in Fig. 3d, e, B$_{D2HG}$-0 responded to D-2-HG supplementation in a dose-dependent manner, and the limit of detection (LOD) and linear range of B$_{D2HG}$-0 were 0.50 and 2–50 μM, respectively (Supplementary Table 4). B$_{D2HG}$-0 exhibited good specificity and did not respond to D-2-HG analogs (L-2-HG, D-lactate, L-lactate, D-malate, and glutarate) or metabolites involved in the tricarboxylic acid cycle (pyruvate, citrate, *cis*-aconitate, isocitrate, 2-KG, succinate, fumarate, L-malate, and oxaloacetate). In addition, a mixture of these chemicals did not interfere with the response of B$_{D2HG}$-0 to D-2-HG (Fig. 3f).

**Sensitivity tuning through binding site mutation**. Reducing the affinity between DhdR and its binding site could increase the equilibrium dissociation constant $K_D$ and finally improve the sensitivity and linear detection range of the biosensor[25]. Thirteen mutations were generated by displacing one or two bases in the hyphenated dyad symmetry of the DhdR binding site (DBS). ITC was used to assess the $K_D$ values of DhdR for the thirteen mutated DBSs (Supplementary Fig. 13). As shown in Table 1, the $K_D$ values of DhdR for the DBS mutants increased with the number of mutated bases. The $K_D$ values of DhdR for D1 and D2 were relatively higher than those of mutants with one mutated base; the $K_D$ values of DhdR for D7, D8, and D12 were relatively higher than those of mutants with two mutated bases (Table 1). These five mutants were then used to construct the biosensors B$_{D2HG}$-1, B$_{D2HG}$-2, B$_{D2HG}$-7, B$_{D2HG}$-8, and B$_{D2HG}$-12, respectively.

As shown in Fig. 4a–e, the five biosensors exhibited different responses to D-2-HG. Increased $K_D$ values of DhdR for the DBS through point mutation could decrease the EC$_{50}$ (concentration of D-2-HG that produces a 50% signal reduction) of the biosensors (Supplementary Table 4). However, it is probably not most suitable for the construction of biosensors when the $K_D$

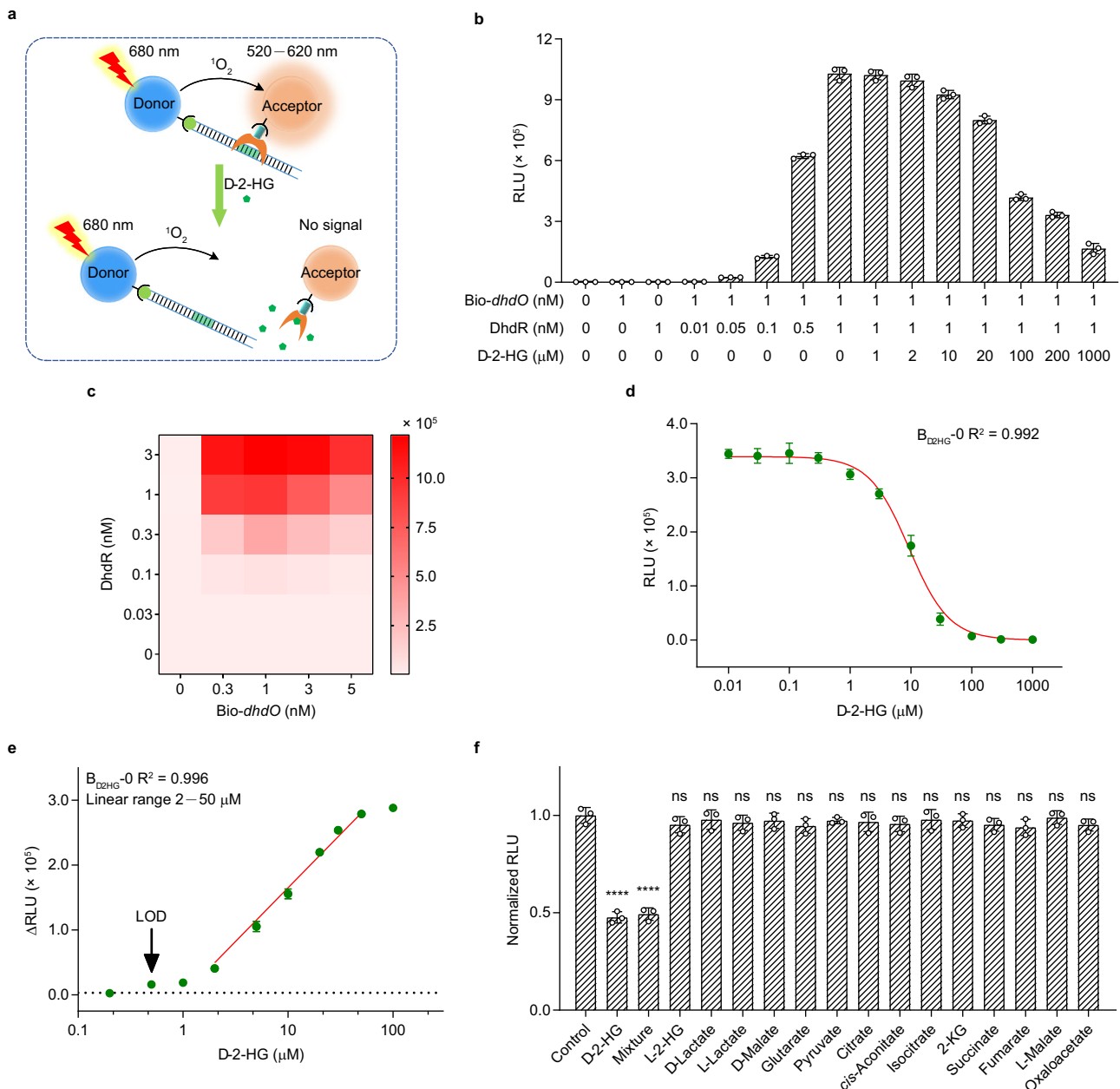

**Fig. 3 Design of the D-2-HG biosensor (B$_{D2HG}$). a** Schematic diagram of the biosensor based on DhdR and AlphaScreen. **b** Evaluation of allosteric effect of D-2-HG on Bio-*dhdO*-DhdR interaction by AlphaScreen. Data shown are mean ± s.d. ($n = 3$ independent experiments). **c** Determination of the optimal concentrations of DhdR and Bio-*dhdO* by cross-titration. Each heat map cell shows the average of the luminescence signals of each titration ($n = 3$ independent experiments). **d** Response of B$_{D2HG}$-0 to different concentrations of D-2-HG. **e** The linear detection range of B$_{D2HG}$-0. Black dotted line is a reference line where the reduced luminescence signal (ΔRLU) is three times of the background signal. **f** Specificity of B$_{D2HG}$-0. The concentrations of tested chemicals were 10 μM. The column mixture contained 10 μM D-2-HG and 10 μM all other tested chemicals. The corresponding luminescence signals are normalized to the maximal value (luminescence signals of control) in the absence of D-2-HG. The concentrations of DhdR and Bio-*dhdO* in **d**–**f** were 0.3 and 1 nM, respectively. Data shown are mean ± s.d. ($n = 3$ independent experiments). The significance was analyzed by a two-tailed, unpaired *t*-test. ****$P < 0.0001$; ns, no significant difference ($P \geq 0.05$). Exact $P$ values are reported in the Source Data file. Source data are provided as a Source Data file.

value of DhdR for DBS is too large. The S/N would reduce and influence the LOD of the biosensor due to the low affinity of DhdR for DBS. Among the five optimized biosensors, B$_{D2HG}$-1 exhibited the lowest LOD (0.10 μM) and a wide linear range (0.3–20 μM) (Fig. 4a–f, Supplementary Fig. 14, and Supplementary Table 4). Thus, this biosensor was selected for the quantification of D-2-HG levels in subsequent experiments.

**Performance of B$_{D2HG}$-1 in D-2-HG quantitation.** The concentration of D-2-HG in body fluids may potentially be used as a

clinical indicator of D-2-HG-related diseases[1,3,32]. Considering the high sensitivity of B$_{D2HG}$-1 for D-2-HG, small amounts of body fluids are required for D-2-HG quantitation. Thus, the dose-response curves of D-2-HG were determined in 100-fold diluted human serum or urine (Fig. 5a, b). Then, D-2-HG, at a range of concentrations, was spiked into human serum or urine and quantified using B$_{D2HG}$-1 and LC-MS/MS. The agreement between the results from LC-MS/MS and the D-2-HG biosensor was satisfactory (Fig. 5c, d). 2-KG, L-glutamate, L-glutamine, and isocitrate are possible precursors of D-2-HG in vivo. Therefore,

these metabolites and L-2-HG (the enantiomer of D-2-HG) were also added to the assay system. As shown in Fig. 5e, f, 2-KG, L-glutamate, L-glutamine, isocitrate, and L-2-HG did not interfere with D-2-HG quantitation in serum or urine.

The D-2-HG biosensor was also applied to quantify the generation of D-2-HG in different cell lines. The dose-response curve for D-2-HG was determined in 10-fold diluted cell culture medium in the absence or presence of L-2-HG, and the biosensor $B_{D2HG}$-1 also produced highly accurate quantification of D-2-HG (Fig. 5g, h and Supplementary Fig. 15). As shown in Fig. 5i, the D-

2-HG biosensor $B_{D2HG}$-1 could detect the increase in D-2-HG levels in the cell culture medium during the growth of HEK293FT cells transfected with *IDH1*/R132H and of HT1080 cells, a fibrosarcoma cell line carrying an *IDH1*/R132C mutation[33]. No D-2-HG was detected in the cell culture medium of HEK293FT cells. GSK864 and AGI-6780 are small molecules that selectively inhibit the cancer-associated mutants of IDH1 and IDH2/R140Q, respectively[34,35]. As expected, treatment with 0.5 μM GSK864 significantly reduced the amount of extracellular D-2-HG in the cultures of HEK293FT-IDH1/R132H and HT1080 cells, while treatment with AGI-6780 did not have any detectable effect (Fig. 5i).

**Table 1 Influence of point mutations in DBS on affinity between *dhdO* and DhdR.**

| No. | Binding sites (5'—3')[a] | $K_D$ (μM)[b] |
|---|---|---|
| *dhdO* 0# | GTTATCAGATAAC | 0.64 ± 0.10 |
| *dhdO* 1# | **T**TTATCAGATAAC | 3.29 ± 0.72 |
| *dhdO* 2# | G**G**TATCAGATAAC | 2.67 ± 0.53 |
| *dhdO* 3# | GT**G**ATCAGATAAC | 1.19 ± 0.25 |
| *dhdO* 4# | GTT**C**TCAGATAAC | 0.58 ± 0.09 |
| *dhdO* 5# | GTTA**G**CAGATAAC | 1.14 ± 0.37 |
| *dhdO* 6# | GTTAT**A**AGATAAC | 0.73 ± 0.18 |
| *dhdO* 7# | **TG**TATCAGATAAC | 5.81 ± 2.15 |
| *dhdO* 8# | **T**T**G**ATCAGATAAC | 4.02 ± 0.92 |
| *dhdO* 9# | **T**TT**C**TCAGATAAC | 3.23 ± 1.27 |
| *dhdO* 10# | **T**TTA**G**CAGATAAC | n.d.[c] |
| *dhdO* 11# | **T**TTAT**A**AGATAAC | 3.23 ± 1.41 |
| *dhdO* 12# | G**GG**ATCAGATAAC | 6.25 ± 1.82 |
| *dhdO* 13# | G**G**T**C**TCAGATAAC | 2.46 ± 0.51 |

[a]The point mutation is shown in bold letter. The 1-bp interval in the palindrome sequence is shown in underlined letter.
[b]The $K_D$ value was determined by ITC and analyzed by using a single-site binding model based on the raw data in Supplementary Fig. 13.
[c]n.d. not detected.

**D-2-HG metabolism in LPS synthesis in *P. aeruginosa* PAO1.** Finally, the D-2-HG biosensor was used to investigate D-2-HG metabolism in *P. aeruginosa* PAO1, an important opportunistic pathogen that causes serious infections[36]. The dose-response curves of D-2-HG in bacterial minimal medium, the growth, and the D-2-HG production of *P. aeruginosa* PAO1 and its derivative were determined (Fig. 6a and Supplementary Figs. 16 and 17). In *P. aeruginosa* PAO1, SerA and D2HGDH (PA0317) are considered the key enzymes responsible for D-2-HG anabolism and catabolism, respectively. Deletion of *PA0317* causes extracellular D-2-HG accumulation in *P. aeruginosa* PAO1 (Δ*PA0317*)[7]. However, quantification of D-2-HG with the biosensor indicated that *P. aeruginosa* PAO1 (Δ*PA0317*Δ*serA*) still accumulated D-2-HG to a certain extent.

Lipopolysaccharide (LPS) functions as a natural barrier against antibiotics and plays a key role in the host-pathogen interactions of *P. aeruginosa*[37]. The synthesis of LPS in *P. aeruginosa* PAO1 involves the dehydrogenation of UDP-GlcNAcA to UDP-2-acetamido-2-deoxy-D-*ribo*-hex-3-uluronic acid [UDP-GlcNAc(3keto)A], which is catalyzed by the UDP-GlcNAcA 3-dehydrogenase, WbpB (Fig. 6b). As shown in Fig. 6a,

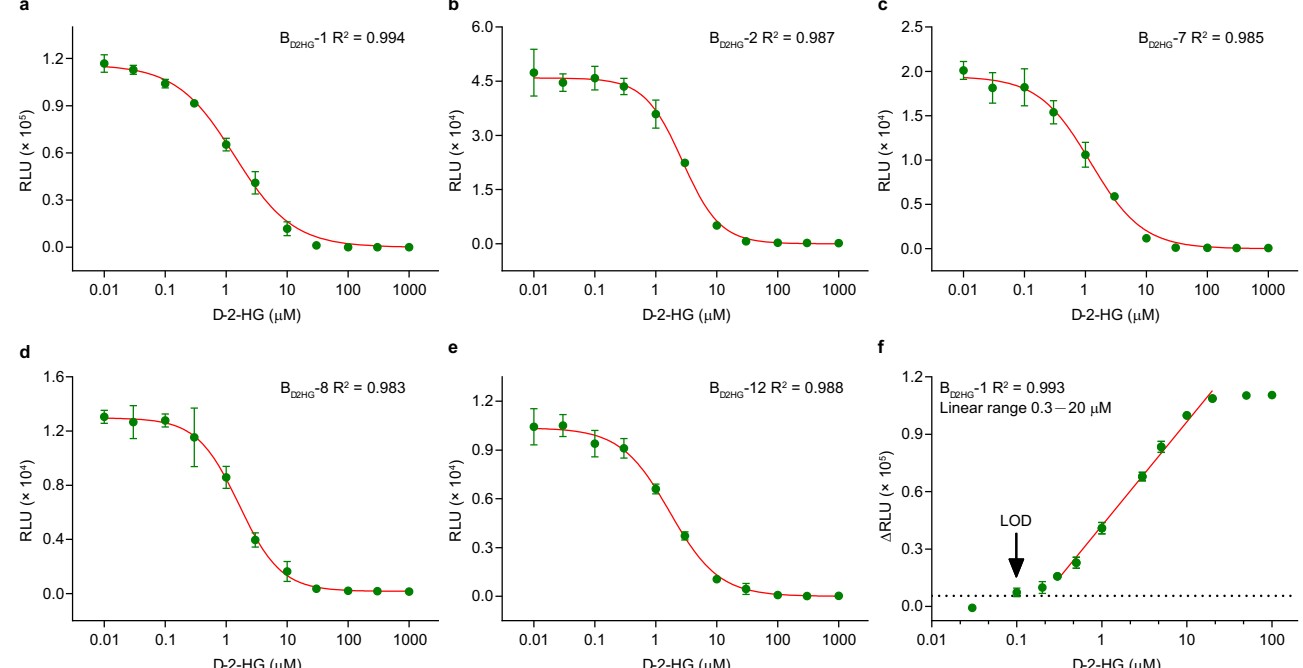

**Fig. 4 Optimization of the D-2-HG biosensor ($B_{D2HG}$). a–e** Responses of the optimized biosensors to different concentrations of D-2-HG. **a** Dose-response curve of $B_{D2HG}$-1; **b** dose-response curve of $B_{D2HG}$-2; **c** dose-response curve of $B_{D2HG}$-7; **d** dose-response curve of $B_{D2HG}$-8; **e** dose-response curve of $B_{D2HG}$-12. **f** The linear detection range of $B_{D2HG}$-1. Black dotted line is a reference line where the reduced luminescence signal (ΔRLU) is three times of the background signal. The concentrations of DhdR and Bio-*dhdO* mutants in **a–f** were 0.3 and 1 nM, respectively. Data shown are mean ± s.d. ($n = 3$ independent experiments). Source data are provided as a Source Data file.

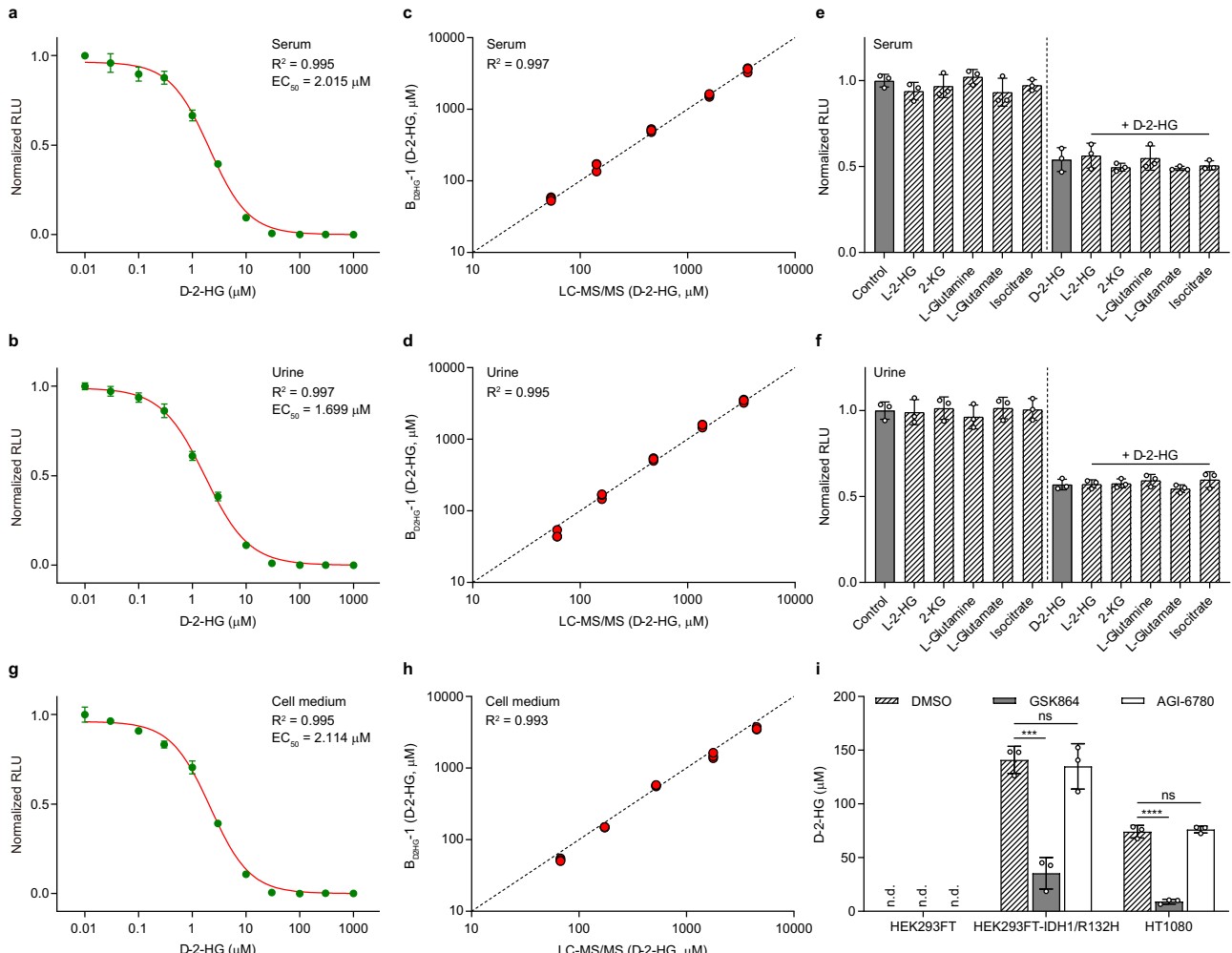

**Fig. 5 Measuring D-2-HG concentrations in various biological samples by using B$_{D2HG}$-1. a** Normalized dose-response curve of B$_{D2HG}$-1 in 100-fold diluted human serum. **b** Normalized dose-response curve of B$_{D2HG}$-1 in 100-fold diluted urine. **c** Comparisons of analysis of D-2-HG in human serum by LC-MS/MS and B$_{D2HG}$-1. **d** Comparisons of analysis of D-2-HG in urine by LC-MS/MS and B$_{D2HG}$-1. Black dotted line indicates a reference line. **e** Effect of possible precursors and enantiomer of D-2-HG on the detection of D-2-HG in serum by B$_{D2HG}$-1. **f** Effect of possible precursors and enantiomer of D-2-HG on the detection of D-2-HG in urine by B$_{D2HG}$-1. **e, f** Metabolites were added in the assay systems with or without D-2-HG at physiologic concentrations (D-2-HG 1 mM, L-2-HG 1 mM, 2-KG 100 μM, L-glutamine 4 mM, L-glutamate 4 mM, and isocitrate 50 μM). **g** Normalized dose-response curve of B$_{D2HG}$-1 in 10-fold diluted cell culture medium. **h** Comparisons of analysis of D-2-HG in cell medium by LC-MS/MS and B$_{D2HG}$-1. Black dotted line indicates a reference line. **i** D-2-HG detection in cell culture medium of HEK293FT cells, HEK293FT-IDH1/R132H cells, and HT1080 cells upon treatment with 0.5 μM GSK864 or AGI-6780, respectively. DMSO was added as the control. The corresponding luminescence signals are normalized to the maximal value in the absence of D-2-HG. n.d., not detected. The concentrations of DhdR and Bio-dhdO-1 were 0.3 and 1 nM, respectively. Data shown are mean ± s.d. (n = 3 independent experiments). The significance was analyzed by a two-tailed, unpaired t-test. \*\*\*P < 0.001; \*\*\*\*P < 0.0001; ns, no significant difference (P ≥ 0.05). Exact P values are reported in the Source Data file. Source data are provided as a Source Data file.

P. aeruginosa PAO1 (ΔPA0317ΔserAΔwbpB) no longer accumulated D-2-HG, suggesting that WbpB may also be involved in D-2-HG anabolism in P. aeruginosa PAO1.

Thus, WbpB of P. aeruginosa PAO1 was overexpressed and purified (Supplementary Fig. 18). The spectral signal peak at 334 nm of the purified WbpB indicates that this enzyme binds NADH tightly (Supplementary Fig. 19). When 2-KG was added to a solution with purified WbpB, the tightly bound NADH was oxidized and the absorbance at 334 nm dramatically decreased (Supplementary Fig. 19). HPLC analysis combined with a luminescence assay using the biosensor B$_{D2HG}$-1 confirmed that WbpB also catalyzes the reduction of 2-KG to produce D-2-HG (Fig. 6c, d). As expected, O-antigen polymers of LPS were completely absent when wbpB was disrupted in P. aeruginosa PAO1 (ΔwbpB) (Fig. 6e and Supplementary Fig. 20). Interestingly, these components clearly decreased in P. aeruginosa PAO1

(ΔPA0317), indicating that D-2-HG catabolism is also involved in LPS synthesis. The reduction in LPS synthesis in P. aeruginosa PAO1 (ΔPA0317) may be caused by the inhibition of WbpB activity through the accumulation of D-2-HG (Fig. 6f). In addition to the biosynthesis of L-serine, D-2-HG metabolism is also involved in the generation of LPS in P. aeruginosa PAO1 (Fig. 6g).

## Discussion

D-2-HG can inhibit the activities of various 2-KG-dependent dioxygenases, transhydrogenase, and transaminase, and influences metabolic homeostasis, cell differentiation, and epigenetics[7,38–41]. However, the mechanism, by which organisms respond to D-2-HG and regulate its catabolism, has not yet been elucidated. To address this, we took advantage of comparative genomics to identify a specific D-2-HG responding regulator

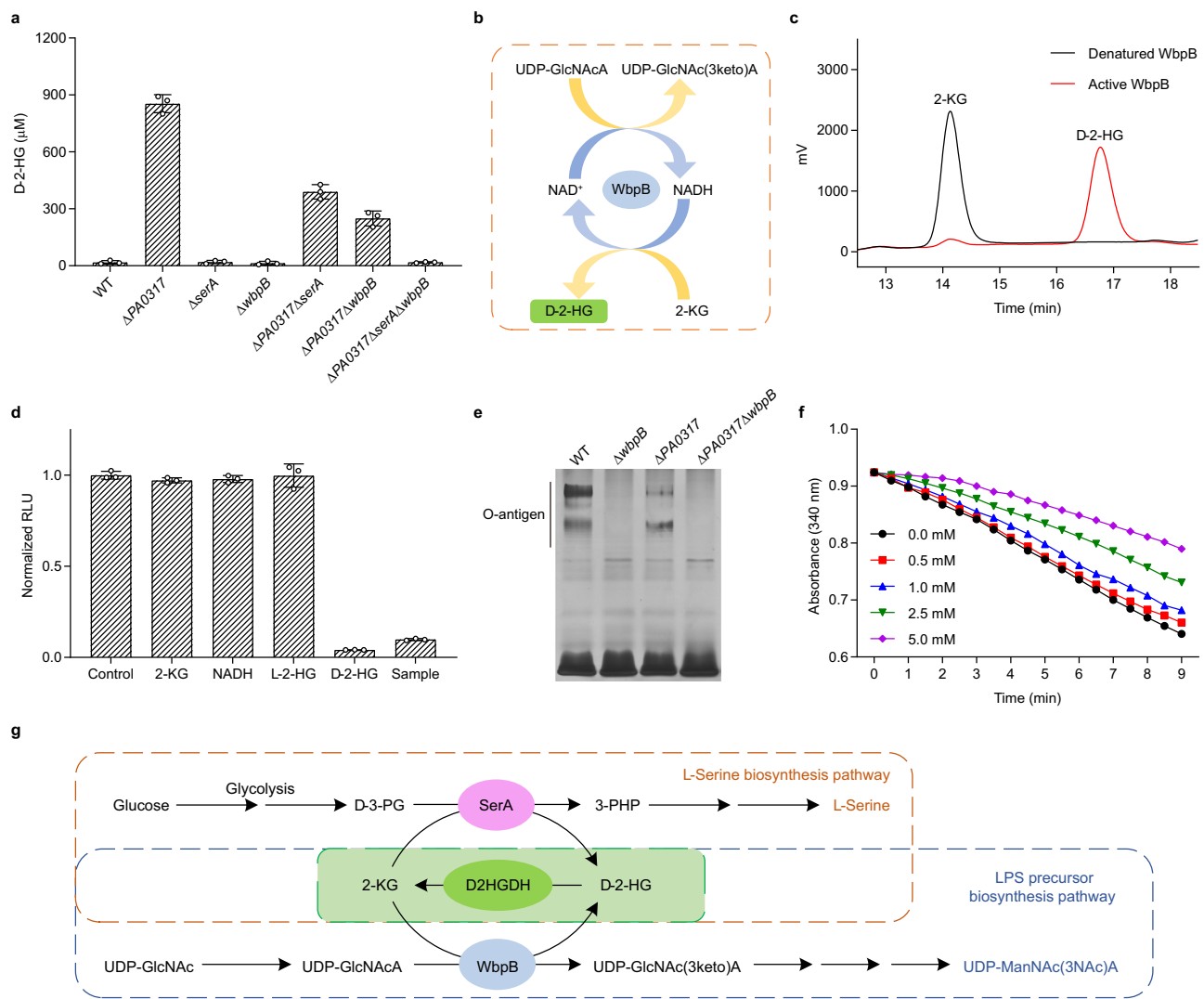

**Fig. 6 Identification of the D-2-HG metabolism pathways in *P. aeruginosa* PAO1 using B$_{D2HG}$-1. a** Determination of extracellular D-2-HG of *P. aeruginosa* PAO1 and its derivatives by B$_{D2HG}$-1. The concentrations of DhdR and Bio-*dhdO*-1 are 0.3 and 1 nM, respectively. Data shown are mean ± s.d. ($n = 3$ independent experiments). **b** Schematic diagram of WbpB-catalyzed coupled reaction. **c** HPLC analysis of the product of WbpB-catalyzed 2-KG reduction. The reaction mixtures containing 2-KG (5 mM), NADH (5 mM), and active or denatured WbpB (0.4 mg mL$^{-1}$) in 50 mM Tris-HCl (100 mM NaCl, pH 7.4) were incubated at 37 °C for 3 h. Black line, the reaction with denatured WbpB; red line, the reaction with active WbpB. **d** Chiral analysis of WbpB produced 2-HG by B$_{D2HG}$-1. The concentration of 2-KG, NADH, L-2-HG, and D-2-HG is 20 μM. The concentrations of DhdR and Bio-*dhdO*-1 were 0.3 and 1 nM, respectively. The sample is 50-fold diluted WbpB-catalyzed product. Data shown are mean ± s.d. ($n = 3$ independent experiments). **e** SDS-PAGE analysis of LPS of *P. aeruginosa* PAO1 and its derivatives. **f** Inhibitory of D-2-HG toward WbpB. The reaction mixtures containing WbpB (0.3 mg mL$^{-1}$), 2-KG (0.5 mM), NADH (200 μM), and 0–5 mM D-2-HG in 50 mM Tris-HCl (100 mM NaCl, pH 7.4) were incubated at 37 °C for 9 min. **g** D-2-HG metabolism pathways in *P. aeruginosa* PAO1. The coupling between D2HGDH and SerA or D2HGDH and WbpB makes the robust interconversion between D-2-HG and 2-KG, and facilitates the biosynthesis of L-serine (orange dotted box) and precursor of LPS in *P. aeruginosa* PAO1 (blue dotted box). Source data are provided as a Source Data file.

DhdR in *A. denitrificans* NBRC 15125. We demonstrated that DhdR plays a critical role in controlling the expression of two co-transcribed genes, *dhdR* and *d2hgdh*. EMSA and DNase I footprinting revealed that DhdR directly interacts with a 13-bp inverted repeat in the *dhdR-d2hgdh* promoter region. D-2-HG acts as an antagonist and can inhibit the binding of DhdR to the promoter. The dissociation between DhdR and the promoter region induced by D-2-HG is essential to initiate the transcription of *d2hgdh* for D-2-HG catabolism (Fig. 2i).

Apart from being an oncometabolite, D-2-HG is also an intermediate of various metabolic processes. For example, we previously characterized D-2-HG metabolism in L-serine biosynthesis in *P. stutzeri* A1501[7]. D-2-HG is endogenously produced by SerA to support the thermodynamically unfavorable D-3-phosphoglycerate dehydrogenation in L-serine biosynthesis. D2HGDH then converts D-2-HG back to 2-KG to prevent the toxic effects associated with the accumulation of D-2-HG. Interestingly, SerA in *A. denitrificans* NBRC 15125 also possesses D-2-HG-producing activity, but SerA-catalyzed endogenous D-2-HG generation does not occur during the growth of this strain. Further investigation confirmed that *A. denitrificans* NBRC 15125 (Δ*serA*) did not exhibit L-serine auxotrophy. SerA-initiated L-serine biosynthesis is a metabolic process essential for the survival of most organisms. Thus, there might be an unidentified pathway for L-serine biosynthesis in *A. denitrificans* NBRC 15125, which should be further investigated.

Elevated D-2-HG levels are considered a biomarker of tumor-associated *IDH1/2* mutations and D-2-HGA[1,2]. Conventional methods for the detection of D-2-HG are GC-MS/MS and LS-MS/MS. Recently, enzymatically-coupled fluorescence assays were developed to measure the concentration of D-2-HG[15,42]. NADH produced during dehydrogenation of D-2-HG by the SerA or NAD-dependent D-2-HG dehydrogenase can be oxidized by the diaphorase to produce fluorescent resorufin. LODs of the enzymatic assays using SerA and NAD-dependent D-2-HG dehydrogenase are 4 and 0.13 µM, respectively. A bioluminescence technique based on NAD-dependent D-2-HG dehydrogenase, NADH:FMN-oxidoreductase, and luciferase has also been developed for the in situ detection of D-2-HG[43]. This technique allows microscopical determination of D-2-HG in sections of snap-frozen tissue with a detection range of 0–10 µmol g$^{-1}$ tissue (wet weight). In this study, DhdR from *A. denitrificans* NBRC 15125, the regulator that specifically responds to D-2-HG, was used as the sensing element to develop a D-2-HG biosensor based on the AlphaScreen technology. Compared with the standard MS-based analytical approaches, quantification of D-2-HG using B$_{D2HG}$-1 is more cost-efficient and can be conducted in 384-well plates for high-throughput detection (Supplementary Table 5). An optimized biosensor, B$_{D2HG}$-1, with high sensitivity (LOD = 0.10 µM) and accuracy (Supplementary Tables 4 and 6), was generated after introducing 13 point mutations in the DBS. Human serum spiked with target determinands is a commonly used alternative for the simulation of clinical samples. For example, commercial human serum spiked with uric acid was used for the evaluation of the performance of uric acid biosensors constructed based on the transcriptional regulator HucR in clinical samples[26]. In this study, the concentration of D-2-HG spiked into serum and urine, and the amount of D-2-HG accumulated in different cell lines could be readily determined using B$_{D2HG}$-1. The results were highly concordant with those of LC-MS/MS. Based on the sensitivity of B$_{D2HG}$-1 and D-2-HG levels in patients, only 1 µL of serum or urine, with no complicated sample pretreatment, might be required for clinical diagnosis of D-2-HG-related diseases.

Biosensors are powerful diagnostic devices that combine a sensing element with specificity for a given analyte and a transducer element that creates a measurable signal for analysis. In addition to AlphaScreen technology, other strategies, such as aTF-based nicked DNA-template-assisted signal transduction (aTF-NAST)[26], CRISPR-Cas12a and aTF mediated small molecule detector (CaT-SMelor)[27], and quantum-dot Förster resonance energy transfer (FRET)[28], can also be coupled with DhdR to construct D-2-HG biosensors. FRET sensors and single fluorescent protein sensors with specific regulators have been widely used for the real-time monitoring of the intracellular levels of various metabolites[44–47]. D-2-HG-sensing fluorescent reporters using DhdR as the sensing element may be developed and used in real-time imaging of intracellular D-2-HG production. Beyond reducing 2-KG to D-2-HG, in many organisms, SerA can also reduce oxaloacetate to D-malate[14]. Given the importance of L-serine in metabolism, D2HGDH in these organisms has evolved to be a versatile enzyme capable of dehydrogenating both D-malate and D-2-HG to eliminate these two metabolites[48]. Interestingly, D2HGDH in *A. denitrificans* NBRC 15125 has no D-malate dehydrogenation activity and only participates in exogenous D-2-HG catabolism. This enzyme with high specificity toward D-2-HG may also be used as a promising sensing element for the development of enzymatic or electrochemical D-2-HG biosensors.

In addition to quantifying D-2-HG in serum and urine, biosensor B$_{D2HG}$-1 was also used to study the mechanism of bacterial D-2-HG metabolism. The findings revealed that D-2-HG metabolism is involved in L-serine biosynthesis and the generation of the O-antigen of LPS in *P. aeruginosa* PAO1. LPS is an important virulence factor of *P. aeruginosa*[37]. Mutant *P. aeruginosa* that is deficient in O-antigen biosynthesis exhibited decreased virulence and showed sensitivity to serum[49]. On the other hand, antibiotic resistance in *P. aeruginosa* is a serious threat to human health[50]. AG-221 and GSK321 target mutant IDHs and have been developed to treat D-2-HG-related cancers[51,52]. Inhibitors targeting the D-2-HG-producing enzyme, WbpB, which was identified in this study, might also be developed as narrow-spectrum antibiotics against *P. aeruginosa*. Inactivation of SerA in *P. aeruginosa* significantly inhibits ExoS secretion and penetration through Caco-2 cell monolayers[53], suggesting that L-serine biosynthesis may also be associated with the pathogenicity of *P. aeruginosa*. Allosteric inhibitors of human SerA have been successfully used to inhibit L-serine biosynthesis in cancer cells and reduce tumor growth[54]. These chemicals may also be effective inhibitors of SerA in *P. aeruginosa* and could potentially be applied as therapies to treat *P. aeruginosa* infection, which is worthy of further investigation.

In summary, we uncovered the regulatory mechanism of D-2-HG catabolism in *A. denitrificans* NBRC 15125, identified a responsive regulator DhdR, and developed a sensitive biosensor for D-2-HG detection. DhdR is a D-2-HG-responding regulator that negatively regulates D2HGDH expression and D-2-HG catabolism. Based on this regulator, we designed and optimized biosensors that can quantify the D-2-HG concentrations in serum, urine, and cell culture mediums. Using the biosensor B$_{D2HG}$-1, which is based on DhdR and AlphaScreen, we also identified the role of D-2-HG metabolism in LPS biosynthesis in *P. aeruginosa* PAO1. The D-2-HG biosensor is thus a valuable experimental tool for revealing other unidentified D-2-HG-related metabolic processes. From a drug-discovery perspective, the fluorometric biosensor with sensitive quantitative capacity may also be potentially utilized in a high-throughput manner. We expect that the D-2-HG biosensor, like other metabolite biosensors[55–57], can be used in screening compounds, mutations, or siRNA libraries to identify new targets that reduce D-2-HG accumulation for therapeutic purposes.

## Methods

**Ethical statement**. This research complies with all relevant ethical regulations. The study protocol was approved by the Research Ethics Committee of the Second Hospital of Shandong University.

**Bioinformatics analysis**. The distribution of D2HGDH in bacteria was checked by using BLAST 2.2.31[7]. Individual genomes showing query coverages more than 90%, E values lower than e$^{-30}$, and maximum identity levels higher than 30% with the query protein (D2HGDH in *P. stutzeri* A1501) were selected for further analysis of the upstream and downstream genes of *d2hgdh*.

**Reagents**. D-2-Hydroxyglutarate (D-2-HG), L-2-hydroxyglutarate (L-2-HG), D-lactate, L-lactate, D-malate, L-malate, D-tartrate, glutarate, 2-ketoglutarate (2-KG), D-glycerate, L-glycerate, succinate, pyruvate, oxaloacetate, *cis*-aconitate, citrate, isocitrate, dichlorophenol-indophenol (DCPIP), 3-(4,5-dimethylthiazol-2-yl)-2,5-diphenyltetrazolium bromide (MTT), phenazine methosulfate (PMS), NADH, NAD$^+$, SYPRO orange, bovine serum albumin (BSA), GSK864, and AGI-6780 used in this study were purchased from Sigma-Aldrich (USA). AlphaScreen donor and acceptor beads were purchased from PerkinElmer (USA). Fumarate and human serum were purchased from Beijing Solarbio Science & Technology Co., Ltd (China). Urine samples were collected from a healthy volunteer; informed consent was provided. D,L-2-Hydroxyglutarate disodium salt (2,3,3-D3) was purchased from Cambridge Isotope Laboratories, Inc. (USA). All other chemicals were of analytical reagent grade.

**Bacteria and culture conditions**. Bacterial strains used in this study are listed in Supplementary Data 1. *A. denitrificans* NBRC 15125 and its derivatives were cultured at 30 °C and 200 rpm. *E. coli*, *P. aeruginosa* PAO1, and their derivatives were cultured at 37 °C and 180 rpm. Luria-Bertani (LB) medium or minimal medium[58]

containing different carbon sources were used to culture bacterial strains. If necessary, antibiotics were added at appropriate concentrations.

**Mammalian cell culture and treatment by IDH mutant inhibitors.** HEK293FT cells were cultured in high-glucose Dulbecco's modified Eagle's medium (MACGENE, China) with 10% (v/v) fetal bovine serum (FBS) (Biological Industries (BioInd)) and 1% penicillin/streptomycin (MACGENE, China). HT1080 cells, obtained from Procell Life Science & Technology Co., Ltd, were cultured in Minimum Essential Medium (MEM) supplemented with 10% (v/v) FBS (ThermoFisher, USA) and 1% (v/v) penicillin/streptomycin (MACGENE, China).

The *IDH1/R132H* gene was synthesized by Sangon Biotech (China) Co., Ltd, and cloned into the lentiviral vector pLenti PGK GFP Puro (w509-5), replacing the EGFP-encoding gene. The constructed vector was named as PGK-*IDH1*/R132H-Puro. HEK293FT packaging cells were co-transfected with PGK-*IDH1*/R132H-Puro, pMD2.G, and psPAX2 by Lipofectamine 3000 (Invitrogen, USA). After 72 h transfection, the supernatant was harvested and filtered through a 0.45 μm filter, and then stored at −80 °C until use. For stable overexpression of IDH1/R132H, HEK293FT cells were infected by the above viral supernatant, cultured for further 2 days, and then selected several rounds with 4–6 μg mL$^{-1}$ puromycin. The obtained cells were called as HEK293FT-IDH1/R132H.

GSK864 and AGI-6780 were dissolved in DMSO. HT1080 ($3 \times 10^5$), HEK293FT ($7 \times 10^5$), and HEK293FT-IDH1/R132H ($7 \times 10^5$) cells were seeded into 6-well plates with 2 mL medium per well, respectively. The medium was changed with fresh medium containing 0.5 μM GSK864 or AGI-6780 the next day. After 48 h, the medium was centrifuged at $13,000 \times g$ for 5 min, and the obtained supernatant was stored at −80 °C until detection.

**Construction of *A. denitrificans* NBRC 15125 and *P. aeruginosa* PAO1 mutants.** The plasmids and primers used in this study are listed in Supplementary Data 1 and 2, respectively. To construct the *A. denitrificans* NBRC 15125 (Δ*d2hgdh*), the homologous arms upstream and downstream of the *d2hgdh* gene were amplified with primers *d2hgdh*-uf/*d2hgdh*-ur and *d2hgdh*-df/*d2hgdh*-dr, respectively. The upstream and downstream fragments were fused together by recombinant PCR with primers *d2hgdh*-uf and *d2hgdh*-dr. The generated fusion fragments were cloned into *Eco*RI and *Bam*HI restriction sites of a suicide plasmid pK18*mobsacB*-tet[59] by using In-Fusion HD Cloning Kit (Clontech, USA) according to the instructions of manufacturer. The generated plasmid, pK18*mobsacB*-*d2hgdh'*, was transferred into *A. denitrificans* NBRC 15125 by the tri-parental mating method in the presence of the helper strain *E. coli* HB101 carrying plasmid pRK2013[60]. The single-crossover mutants with the integration of the plasmid pK18*mobsacB*-*d2hgdh'* into the chromosome were screened from M9 medium plates containing 20 g L$^{-1}$ trisodium citrate as the sole carbon source and 40 μg mL$^{-1}$ tetracycline. The double-crossover cells were selected on LB plates containing 15% (w/v) sucrose. The *serA* and *dhdR* mutants of *A. denitrificans* NBRC 15125 were constructed by using the same procedure.

*P. aeruginosa* PAO1 (Δ*wbpB*) was constructed based on RedγBAS recombineering system[61]. A DNA fragment containing gentamicin resistance gene flanked by lox71/lox66 sites and 75 bp homologous arms of *wbpB* were amplified with primers *wbpB*-genta-loxM-F/*wbpB*-genta-loxM-R. The PCR products were transferred into the *P. aeruginosa* PAO1 cells, which harbored plasmid pBBR1-Rha-redyBAS-kan and were induced by L-rhamnose. After gentamicin selection (30 μg mL$^{-1}$), the obtained gentamicin-resistant strain was further transformed by the plasmid pCM157. Then, the right colonies were screened from LB plates containing 60 μg mL$^{-1}$ tetracycline, and treated by isopropyl β-D-thiogalactoside (IPTG) induction to eliminate the gentamycin resistance, resulting in the *wbpB* mutant strain *P. aeruginosa* PAO1 (Δ*wbpB*). The *serA* mutant of *P. aeruginosa* PAO1 was constructed by using the same procedure. All plasmids and fragments were transferred by electroporation. All the constructed mutants were confirmed by PCR and sequencing.

**Expression, purification, and characterization of D2HGDH, SerA, DhdR, and WbpB.** The D2HGDH-encoding gene *d2hgdh* was amplified with primers *d2hgdh*-F/*d2hgdh*-R, which contained *Sac*I and *Hin*dIII restriction enzyme sites, respectively. The PCR product was digested with *Sac*I and *Hin*dIII, and then ligated into the expression vector pETDuet-1. The generated plasmid pETDuet-*d2hgdh* was introduced into *E. coli* BL21(DE3) for D2HGDH expression. To overexpress SerA, DhdR, and WbpB, the corresponding strains were generated by using the same procedure.

The constructed *E. coli* BL21(DE3) strains were grown in LB medium at 37 °C to an OD$_{600nm}$ of 0.6 and induced with 1 mM IPTG at 16 °C for 12 h. The cells were harvested by centrifugation at $6,000 \times g$ for 10 min at 4 °C and washed twice with buffer A (20 mM sodium phosphate, 20 mM imidazole, and 500 mM NaCl, pH 7.4). The cell pellets were then resuspended in buffer A containing 1 mM phenylmethanesulfonyl fluoride (PMSF) and 10% (v/v) glycerol, lysed by sonication, and centrifuged at $13,000 \times g$ for 50 min at 4 °C. The supernatant was loaded onto a 5 mL HisTrap HP column (GE Healthcare, USA) pre-equilibrated with buffer A, and then eluted with a gradient ratio of buffer B (20 mM sodium phosphate, 500 mM imidazole, and 500 mM NaCl, pH 7.4) at a flow rate of 5 mL min$^{-1}$. The purified proteins were analyzed by 13% sodium dodecyl sulfate-

polyacrylamide gel electrophoresis (SDS-PAGE). Protein concentrations were measured by using the Bradford protein assay kit (Sangon, China).

To determine the native molecular weight of different proteins, size exclusion chromatography was performed using a Superdex 200 10/300 GL column (GE Healthcare, USA). The column was equilibrated with buffer containing 50 mM sodium phosphate and 150 mM NaCl (pH 7.2) at a flow rate of 0.5 mL min$^{-1}$. Thyroglobulin (669 kDa), ferritin (440 kDa), aldolase (158 kDa), conalbumin (75 kDa), ovalbumin (44 kDa), and RNase A (13.7 kDa) were used as standard proteins.

**Enzymatic assays of D2HGDH and SerA.** *A. denitrificans* NBRC 15125 were cultured to mid-log stage in minimal medium containing different compounds as carbon sources. Cells were harvested, washed twice, resuspended to an OD$_{600nm}$ of 20 in 50 mM Tris-HCl buffer supplemented with 1 mM PMSF and 10% (v/v) glycerol, then lysed by sonication on ice. The homogenate was centrifuged at $13,000 \times g$ for 15 min at 4 °C, and the supernatants (the crude cell extracts) were used for D2HGDH activity measurement.

The activity of D2HGDH was assayed at 30 °C by determining the reduction of DCPIP spectrophotometrically at 600 nm in 800 μL reaction mixture containing 50 mM Tris-HCl buffer (pH 7.4), 0.2 mM PMS, 0.05 mM DCPIP, and variable concentrations of substrate. One unit (U) of D2HGDH activity was defined as the amount of enzyme that reduced 1 μmol of DCPIP per minute. The activity of SerA was assayed at 30 °C by measuring the oxidation of NADH spectrophotometrically at 340 nm in 800 μL reaction mixture containing 50 mM Tris-HCl buffer (pH 7.4), 0.2 mM NADH, and variable concentrations of 2-KG. One unit (U) of SerA activity was defined as the amount of enzyme that oxidized 1 μmol of NADH per minute. The kinetic constants were calculated by using the Lineweaver–Burk plot method. All the assays were performed using a UV/visible spectrophotometer (Ultrospec 2100 pro, Amersham Biosciences, USA).

**Extraction of intracellular D-2-HG.** *A. denitrificans* NBRC 15125 and its derivatives were cultured in LB medium to mid-log phase. The cells were harvested by centrifugation at $3,500 \times g$ for 15 min at −9 °C, and washed with 10 mL ice-prechilled PBS. The intracellular D-2-HG was then extracted by using the boiling ethanol extraction method[62]. The cell pellets were resuspended with 1 mL absolute ethanol and were boiled for 15 min. After being cooled on ice for 5 min, the supernatant was obtained by centrifugation at $13,000 \times g$ for 5 min at 4 °C and dried in a vacuum centrifuge (60 °C). The dried residue was then resuspended in Mili-Q water and stored at −80 °C before analysis. The cell volume of 1 mL cell cultures at an OD$_{600nm}$ of 1 was assumed as 3.6 μL[63].

**Analysis of the regulation mechanism of DhdR.** *RNA preparation and RT-PCR.* Total RNA was extracted from *A. denitrificans* NBRC 15125 cells grown in minimal medium containing 3 g L$^{-1}$ D-2-HG or pyruvate as the sole carbon source using the Easypure RNA Kit (Transgen, China). RNA integrity was checked by 1.5% agarose gel electrophoresis. The elimination of genomic DNA contamination and the synthesis of total cDNA was conducted by using HiScript II Q RT SuperMix for qPCR (+gDNA wiper) Kit (Vazyme, China) according to the instructions of the manufacturer. Reverse transcription-PCR (RT-PCR) was performed using the total cDNA as template and *d-d*-RTF/*d-d*-RTR as primers. The amplicon was designed to be a 301-bp fragment and located overlapping with *d2hgdh* and *dhdR* genes.

*Determination of the transcriptional start site.* The transcription start site of *dhdR-d2hgdh* operon was identified by a 5′ rapid amplification of cDNA ends (RACE) system (Invitrogen, China). The first-strand cDNA was synthesized from total RNA with primer GSP1, tailed with terminal deoxynucleotidyl transferase and dCTP, and then amplified by PCR with the abridged anchor primer (APP) and GSP2. A nested PCR was carried out using the PCR product as a template with APP and GSP3. The PCR product was cloned into pMD18-T vector (TaKaRa, China) for sequencing.

*Electrophoretic mobility shift assay (EMSA).* The DNA fragment F1 was generated by PCR from *A. denitrificans* NBRC 15125 genomic DNA using primers F1-F/F1-R. 10 nM DNA was incubated with purified DhdR (0–100 nM) at 30 °C for 30 min in 20 μL EMSA binding buffer [10 mM Tris-HCl, pH 7.4, 50 mM KCl, 0.5 mM ethylenediaminetetraacetic acid (EDTA), 10% (v/v) glycerol, and 1 mM dithiothreitol (DTT)]. The mixtures were then analyzed by electrophoresis with 6% native polyacrylamide at 4 °C and 170 V (constant voltage) for 45 min. The gel was stained with SYBR green I (TaKaRa, China) for 30 min and photographed by G:BOX F3 gel doc system (Syngene, USA). For the effector analysis of DhdR, purified DhdR (70 nM) was incubated with 40 mM D-2-HG, L-2-HG, D-malate, D-lactate, glutarate, 2-KG, or pyruvate at 30 °C for 15 min. The DNA fragment (10 nM) was subsequently added into the mixtures and incubated for another 30 min before electrophoresis.

*DNase I footprinting assay.* DNase I footprinting assay was conducted using 6-carboxyfluorescein (FAM) labeled probe and purified DhdR. The fragment F1 was amplified and inserted into pEASY-Blunt Simple Cloning Vector (TransGen, China) to generate pEASY-Blunt-F1. The FAM-labeled probe was amplified with primers M13F-FAM (priming site: bases 290–305 of pEASY-Blunt Simple Cloning Vector) and M13R (priming site: bases 205–221 of pEASY-Blunt Simple Cloning

Vector) and plasmid pEASY-Blunt-F1 as template. The PCR product was purified after agarose gel electrophoresis, and quantified by NanoDrop 2000C (Thermo Scientific, USA). For each assay, 400 ng FAM-labeled probe was incubated with 1 µg DhdR in 40 µL EMSA binding buffer at 30 °C for 30 min. Next, 10 µL solution containing about 0.015 U DNase I (Promega, USA) and 100 nmol CaCl$_2$ was added and incubated for 25 °C for 1 min. The reaction was terminated by adding 140 µL stop solution [30 mM EDTA, 200 mM sodium acetate, and 0.15% (w/v) SDS]. Digested samples were subsequently extracted with phenol/chloroform, precipitated with ethanol, dissolved in 30 µL Mili-Q water.

*Fluorescence-based thermal shift (FTS) assay.* For FTS assays, 6 µM purified DhdR, 4 × SYPRO orange and 100 µM tested compounds in FTS buffer (20 mM sodium phosphate and 300 mM NaCl, pH 8.0) with a total volume of 25 µL were added in 96-well white PCR plates (Roche, USA). The temperature was increased from 25 to 95 °C at 0.07 °C/s and the fluorescence intensities were read 8 times per °C using the LightCycler 480 system (Roche, USA) with excitation at 465 nm and emission at 580 nm. The Tm was determined according to the derivative melt curves by calculating the first derivative of the raw fluorescence data at each temperature. The control experiment was performed by replacing the ligand with FTS buffer. The ΔTm values were calculated by subtracting the Tm of control. The compounds which ΔTm value > 2 °C are considered hits and investigated subsequently by isothermal titration calorimetry (ITC) to obtain the definite proof of ligand binding.

**Development of the biosensors.** *ITC experiments.* DhdR, D-2-HG, and the DNA fragments were dissolved in the same buffer as FTS assays. All the DNA fragments were 27 bp in length, obtained by annealing of two reverse complemented primers listed in Supplementary Data 2. Calorimetric assays were conducted on a MicroCal PEAQ-ITC (Malvern, UK) at 25 °C with a stir speed of 750 rpm. The reaction cell was filled with 300 µL DhdR solution, and the titration was performed with an initial 0.4 µL injection of 100 µM DNA fragment, followed by 18 injections of 2 µL by 150 s intervals. The control experiment was performed by titrating D-2-HG or the corresponding DNA fragment into buffer. The net heat of the dilutions was corrected by subtracting the heat of the control point-to-point. The equilibrium dissociation constant ($K_D$) was obtained based on a single-site binding model with the assistance of MicroCal PEAQ-ITC analysis software.

*Biotinylated DNA preparation.* The biotinylated DNA *dhdO* (Bio-*dhdO*) was prepared by two round PCR. First, unlabeled *dhdO* fragment was amplified via recombinant PCR with the primers *dhdO*-F and *dhdO*-R. Then, Bio-*dhdO* fragment was amplified with primers Bio-F and *dhdO*-R and unlabeled *dhdO* as template. The PCR products were purified after agarose gel electrophoresis, and quantified by NanoDrop ND-1000 (Thermo Scientific, USA). The mutants Bio-*dhdO*-1, -2, -7, -8, and -12 were prepared by using the same procedure.

*AlphaScreen assays.* All the AlphaScreen assays were carried out in 384-well white OptiPlates (PerkinElmer, USA) at room temperature. The 25 µL detecting solution contained 5 µL of each component: DhdR, biotinylated DNA fragment, donor beads, acceptor beads, and samples to be measured. DhdR, biotinylated DNA fragment, and samples were mixed and incubated for 30 min. Then, 20 µg mL$^{-1}$ acceptor beads were added and incubated in dark for 30 min. Finally, 20 µg mL$^{-1}$ donor beads were added and incubated for 60 min. The luminescence was measured by EnSight Multimode Plate Reader (PerkinElmer, USA). The dose-response curves were fitted with [Inhibitor] vs. response-Variable slope (four parameters) models of GraphPad Prism 7.0.

*Buffer optimization.* The buffers used in AlphaScreen assays were described as follows. HBS-P buffer contained 10 mM HEPES and 150 mM NaCl; HBS-EP buffer was HBS-P buffer containing additional 3 mM EDTA; AH buffer contained 25 mM HEPES and 100 mM NaCl; PBS-P buffer contained 100 mM phosphate-buffered saline (PBS). All buffers were adjusted to pH 7.4, supplemented with 0.1% (w/v) BSA and 0.005% Tween-20.

**Preparation and analysis of LPS.** LPS was extracted by using Hitchcock and Brown LPS preparation method[64]. Briefly, the cells of *P. aeruginosa* PAO1 and its derivatives grown in LB medium were harvested by centrifugation at 15,600 × *g* for 1 min, washed three times, and resuspended to an OD$_{600nm}$ of 0.45 with PBS. Then, the cell suspension (1 mL) was resuspended in 250 µL lysis buffer [1 M Tris, 10% (v/v) glycerol, and 2% (w/v) SDS, pH 6.8] and heated at 100 °C for 15 min. After adding 30 µg proteinase K and incubating at 55 °C for 5 h, the LPS samples were cooled to room temperature, separated by SDS-PAGE with 13% polyacrylamide gels, and visualized by a silver-staining method. 30% (v/v) ethanol-10% (v/v) acetic acid containing 7 g L$^{-1}$ periodic acid was used to oxidize LPS in the gel at 22 °C for 20 min. The gel was washed three times with ddH$_2$O for 5 min, stained with 1 g L$^{-1}$ AgNO$_3$ for 30 min at 30 °C. Then, the gel was washed with ddH$_2$O for 10 s. The color was developed by 30 g L$^{-1}$ Na$_2$CO$_3$ and 0.02% (v/v) formaldehyde (pre-cooled on ice). After terminating the color development reaction by 10% (v/v) acetic acid and washing with ddH$_2$O, the gel was photographed by CanoScan 9000F MarKII (Cano, Japan).

**Analytical methods.** *High-performance liquid chromatography (HPLC) analysis.* HPLC system LC-20AT (Shimadzu, Japan) was used to identify the catalytic products in enzymatic reaction samples. The samples were deproteinized before HPLC analysis. In brief, the samples were heated at 100 °C for 15 min. The precipitate of denatured protein was removed by centrifugation at 13,000 × *g* for 15 min and was filtered

through a 0.22 µm filter. The deproteinized supernatant was analyzed using an Aminex HPX-87H column (300 × 7.8 mm, Bio-Rad, USA) and a refractive index detector. 10 mM H$_2$SO$_4$ was used as the mobile phase with a flow rate of 0.4 mL min$^{-1}$.

*Liquid chromatography-tandem mass spectrometry (LC-MS/MS) analysis of D-2-HG.* LC-MS/MS system was applied to quantify the D-2-HG in various biological samples. 2,3,3-D3 was supplemented into biological samples as internal standard. Urine samples and cell medium samples were deproteinized with the same procedure used in HPLC analysis; human serum samples were mixed with methanol and vortexed for 2 min, then removed the precipitate of denatured protein as described above. Analysis was performed on an HPLC system LC-20A (Shimadzu, Japan) coupled with a Prominence nano-LTQ Orbitrap velos pro ETD mass spectrometer (ThermoFisher, USA) operating in negative ion mode. The mobile phase consisted of a mixture of methanol and 0.1% triethylamine adjusted to pH 4.5 with acetic acid (5:95). The total run time was 15 min, and 20 µL samples were injected into a Chirobiotic R column (250 × 4.6 mm, Supelco Analytical, USA) with a flow rate of 0.5 mL min$^{-1}$. Monitored transitions for D-2-HG and internal standard were 147.0 → 129.0 m/z and 150.0 → 132.0 m/z, respectively.

**Statistics and reproducibility.** Data analyses were conducted using Microsoft Excel 2016, Graphpad Prism 7 (Graphpad), and OriginPro 2016 (OriginLab). The ITC data were processed using MicroCal PEAQ-ITC analysis software 1.1.0.1262. The imaging data were obtained using CanoScan 9000 F MarKII (Cano) and processed using ImageJ 1.52p. All data shown are means ± s.d. and were analyzed using unpaired two-tailed *t*-test where appropriate; *, $P < 0.05$; **, $P < 0.01$; ***, $P < 0.001$; ****, $P < 0.0001$; ns, no significant difference ($P ≥ 0.05$). For SDS-PAGE analysis, RT-PCR, and EMSAs, similar results were obtained from three independent experiments.

**Reporting summary.** Further information on research design is available in the Nature Research Reporting Summary linked to this article.

## Data availability

The data generated in this study are provided in the Supplementary Information file and the Source Data file. Source data are provided with this paper.

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

## Acknowledgements

This work was supported by the grants of National Key R&D Program of China (2018YFA0901200 to C.M.), National Natural Science Foundation of China (31800025 to W.Z. and 31970055 to C.G.), Shandong Provincial Funds for Distinguished Young Scientists (JQ 201806 to C.G.), and Qilu Young Scholar of Shandong University (to C.G.). The funders had no role in study design, data collection, and interpretation, or the decision to submit the work for publication. We also thank Dr. Zhifeng Li, Dr. Jingyao Qu, and Dr. Jing Zhu from SKLMT (State Key Laboratory of Microbial Technology, Shandong University) for assistance in isothermal titration calorimetry (ITC) and mass spectrographic analysis.

## Author contributions

C.G., C.M., and P.X. designed the research. D.X., W.Z., X.G., Y.L., C.H., S.G., Z.K., and X.X. performed the research. D.X. and C.G. analyzed the data. D.X., W.Z., C.G., C.M., and P.X. wrote the paper.

## Competing interests

The authors declare no competing interests.
