## [Peer Review File · Nature Communications]

Reviewers' Comments:

Reviewer #1:

Remarks to the Author:

A D-2-hydroxyglutarate biosensor based on specific transcriptional regulator Dh dR

Xiao et al report the characterisation a regulatory protein and enzymes involved in the metabolism of D-2-hydroxyglutarate (D2HG). This metabolite has been linked to cancer and is also linked to LPS biosynthesis in bacteria. Initially the identified metabolic genes/enzymes were used to study anabolism and catabolism of D2HG in the bacterial strain *Achromobacter denitrificans*. The associated regulatory protein and the corresponding promoter/operator were then biochemically characterised and used to develop a biosensor based on Alpha screen technology. Various optimisation steps were performed (media, DNA:protein ratio, DNA sequence) to improve biosensor performance. The biosensor was then applied to detect D2HG spiked into serum and other media, plus from mammalian cells producing D2HG and finally from bacterial cells also producing D2HG. The manuscript reports an extensive piece of work from comparative microbial genomics, metabolic pathway analysis, biochemical characterisation through to biosensing application in biomedical/microbiology. From practical/utility perspective it would be beneficial if the authors could highlight the potential benefits (cost, time, portability) of using the biosensor approach compared to standard analytical MS-based approaches. Overall, the experiments seem to be appropriate and well performed, albeit for a few minor queries/errors detailed below.

Minor comments:

General: Protein names used throughout when referring to genes, rather than italicised gene names. Extracellular and intracellular activity used when it was not clear if the authors mean exogenous and endogenous addition/production.

L27 "dehydrogenase" not "dehydrogenases"

L34 amplified luminescent proximity homogeneous assay technology (Alpha Screen).

L49 would be useful to know the physiological relevant concentrations of D2HG upfront

L68 This aTF can repress the...

L69 ...and specifically is derepressed by D-2-HG.

L75 what is D-2-HG generated from?

L82 Genes encoding GntR...

L88 GntR and CdaR - these are proteins, correct to "genes encoding GntR..." or change to italicised gene names

L90, L91, L94, L95, L96 same as above - gene names required

L114 SerA appears to be a tetramer based on calibration curves ~ 160 kDa = $4 * 40$ kDa - may require sec-MALS to resolved accurate multimeric form

L116 enzyme kinetic parameters may need to be revised based on correct multimeric form

L119 intracellular accumulation?

L129 "D2HGDH is critical for extracellular D-2-HG utilization" is D2HGDH secreted? Not clear what is meant here relative to "lack intracellular metabolism" observed for D2HGDH see L122 - please clarify/rephrase this section

L135 was RT-PCR perform in the absence of D2HG this would seem a logical experiment to perform to characterise native regulation/performance?

L135-6 Where is the data to support the TSS?

L136 ...dh dR was identified by rapid amplification of cDNA ends (RACE).

L148 ... region overlaps with the...

L166 "sensing" rather than "biorecognition" maybe a more appropriate term?

L183 remove "extra"

L199 add comment on the upper detection limit and dynamic range

L215 for biosensor / effector terminology EC50 is more commonly used

L217 further expansion of the different outputs would have been interesting here. Lower affinity initially appears to increase output RLU signal i.e. designs 1 & 2 vs 0, but further decreasing in affinity leads a drop off in output - why? Further decreased affinity between the aTF and operator appears to result in more sensitive dose response curves. It would be useful here to understand the desired performance functions i.e. sensitivity, dynamic etc that you seeking to achieved. And

therefore from design rules perspective it would be useful here to aid future biomolecular engineers to how this may be achieved through DNA/promoter engineering etc
L230 ...in vivo, therefore these...
L239 IDH1 - definition required what is this enzyme/gene - also use gene name
L240 gene name
L249 Based on - remove. ...The dose-response curves...
L251 "assayed" replace with "determined"
L333 does D2HGDH have a signal peptide? Clarify what extracellular catabolism means
L495, L499, L517 primer sequence and amplicon locations required
L972 cell line descriptions required here
Fig1d - chemical structures would be useful - see below (Figure 3a)
Fig 1e - re-scaled concentration y-axis
Fig 2a - how is this demonstrated? amplicon location details required - please add
Fig3a - figure panel refers to the regulation of operon - would seem to fit better with the previous figure / section rather than the 'biosensor' work - also include chemical structures.
Supp Figure 9-1 "D1" the isotherm does not seem to correspond to the displayed fitted data below - raw data indicates no binding? check and provide more convincing data

Neil Dixon

Reviewer #2:

Remarks to the Author:

This study from Xiao et al describes the development and application of a novel D-2-HG biosensor based on a bacterial allosteric transcription factor which they name DhdR. They leverage the AlphaScreen approach to create a bead-based assay with high selectivity for D-2-HG and use this to probe in biologically relevant conditions. Further they apply the biosensor to identify a potential role for PAO1 in *Pseudomonas aeruginosa* as a protein that participates in D-2-HG synthesis. I have only extremely minor comments with respect to the work.

- (1) 13 point mutants are generated in order to better optimize the biosensor, yielding a significant decrease in LOD to 0.08 μ M. Is there a rationale for why this point mutation was efficacious?
- (2) The authors demonstrate the biosensor's sensitivity in the context of L-2-HG generation in IDH1 mutant HEK293T cells, though it is unclear how much 2-HG (L v D) is generated in these cells. It is known that the relative levels of L-2-HG can be quite high relative to D-2-HG, sometimes more than 1000-fold with L-2-HG being in the mM. Have the authors tested the ability of the sensor to accurately quantify μ M concentrations of D-2-HG in this context?
- (3) Can the authors comment on the ability to use such an approach as an intracellular biosensor coupled to other technologies for real-time imaging of D-2-HG production?

Reviewer #3:

Remarks to the Author:

This study describes the identification of a transcriptional regulator that recognizes D-2-HG and the development and analysis of a series of derived biosensor. This is an interesting study with significant potential. However, this reviewer has detected several major and minor flaws and inconsistencies, listed below, that have to be addressed

In the introduction the authors state that "However, chiral derivatization with proper reagents is necessary to distinguish D-2-HG from its mirror-image enantiomer L-2-HG, and these methods are time-consuming and laborious." It would be fair to add that both isomers can be probably resolved by ion-mobility qTOF mass spectrometry, without any derivatization.

The authors determined V_{max} of the D-2-HGDH enzyme and provide U / mg as unit. However, $V_{ma} \times E(t)$ (total enzyme concentration), therefore the unit for V_{max} is typically μ M/min. The unit U/mg is the unit for the specific enzyme activity. V_{max} and specific enzyme activity are not the same. This needs to be corrected.

Electrophoretic mobility shift: Although the EMSA data are clear, there were no controls included. Typically, EMSA controls involve the binding to non-specific DNA or competition assays with the specific, unlabeled DNA fragment.

The authors state that D-2-HG is the specific effector of DhdR. However, this statement is based solely on the analysis of 6 other compounds. To consolidate this claim further experiments are required. This referee recommends the use of thermal shift assays of the transcriptional regulator in the presence of compounds from compound libraries such as Biolog compound arrays. Such assays can be conducted in a high-throughput format using 96 well plates.

Isothermal titration calorimetry was used to determine the thermodynamic binding parameters of DhdR. This manuscript would benefit from ITC studies of the binding of D2-HG to DhdR.

From Fig. 3A an I50 value (half maximal signal) of approximately 10 microM D-2-HG was determined. These data appear to contrast with Fig. 3C showing that 10 microM D2-HG caused only minor changes.

A key parameter of any analytical technique is the sensitivity or onset of response, i.e. the lowest concentration at which the presence of a given compound can be detected with confidence. The authors claim that this value is of 0.8 microM for the initial biosensor. However, considering that the I50 is about 10 microM and considering the data shown in Fig. 3e, it is unrealistic to state that the sensitivity of the biosensor is at 0.8 microM.

The biosensor was then improved by mutation the operator site. The dose response for the best biosensor is shown in Fig. 4b and the authors claim that the sensitivity of the system is of 0.08 microM. However, inspection of Fig. 4b shows that at 0.08 microM is in the noise area, particularly taking into account the error associated with the data at 1 microM. In addition, Supp. 10b (of the same optimized biosensor) reveals that the data obtained for 0.1 and 0.5 are basically the same and close to zero (background signal), a fact that is incompatible with the claim that the biosensor has a sensitivity of 0.08 microM.

There is a major flaw in ITC data interpretation. Data are not valid since not corrected for dilution heats prior to curve fitting. In some cases dilution heats are very important.

At line 223 the authors state that the concentration of D2-HG has the potential to be a clinical indicator. It would be of interest to know this concentration in human samples and relate it to the sensitivity of the biosensor. To assess the performance of the biosensor the authors have spiked human serum and urine with D-2-HG and quantified the compound by ms and the developed biosensor. As detailed in the introduction the main motivation of this work is the development of a biosensor of an oncometabolite. The main concern of this reviewer is the lacking proof of concept of this biosensor to detect this oncometabolite. The authors have spiked human serum with D-2-HG and have detected this metabolite in the corresponding samples. However, this is not a proof of concept of this biosensor can be used to detect oncometabolite. To this issue samples from cancer patients and samples from healthy individuals need to be analysed to assess whether the biosensor is indeed able to differentiate these samples.

In addition, a revision of the English is required, in particular the use of articles and the use of tenses: i.e. past tense when reporting data and present tense for discussion and interpretation

Line 44: been reported

Line 68: typically "de-repress" is used

Responses to the reviewers' and editor's comments point-by-point

NCOMMS-21-12376-T

Thanks a lot for the reviewers' comments, which are very useful for us to improve our manuscript. With regard to reviewers' comments and suggestions, we reply as follows:

REVIEWER COMMENTS

Reviewer #1 (Remarks to the Author):

A D-2-hydroxyglutarate biosensor based on specific transcriptional regulator DhdR

Xiao et al report the characterisation a regulatory protein and enzymes involved in the metabolism of D-2-hydroxyglutarate (D2HG). This metabolite has been linked to cancer and is also linked to LPS biosynthesis in bacteria. Initially the identified metabolic genes/enzymes were used to study anabolism and catabolism of D2HG in the bacterial strain Achromobacter denitrificans. The associated regulatory protein and the corresponding promoter/operator were then biochemically characterised and used to develop a biosensor based on Alpha screen technology. Various optimisation steps were performed (media, DNA:protein ratio, DNA sequence) to improve biosensor performance. The biosensor was then applied to detect D2HG spiked into serum and other media, plus from mammalian cells producing D2HG and finally from bacterial cells also producing D2HG. The manuscript reports an extensive piece of work from comparative microbial genomics, metabolic pathway analysis, biochemical characterization through to biosensing application in biomedical/microbiology. From practical/utility perspective it would be beneficial if the authors could highlight the potential benefits (cost, time, portability) of using the biosensor approach compared to standard analytical MS-based approaches. Overall, the experiments seem to be appropriate and well performed, albeit for a few minor queries/errors detailed below.

Response: Thanks for your positive comments. We have made corresponding changes according to your kind suggestions and have added a short sentence in “Discussion” section as

follows:

“Compared with the standard MS-based analytical approaches, quantification of D-2-HG using B_{D2HG-1} is more cost-efficient and can be conducted in 384-well plates for high-throughput detection (Supplementary Table 5).”

Supplementary Table 5 Comparison of LC-MS/MS and B_{D2HG}-1 of D-2-HG detection.

Method	Cost (\$/reaction)	Sample preparation	Availability of microplate	Samples/Detection ^c	Ref.
LC-MS/MS	> 54.25 ^a	Deproteinization required	No	1	This study
B _{D2HG} -1	> 3.66 ^b	No sample pre-treatment	Yes, 384-well plates	384	This study

^aCalibrated based on the quoted price for sample testing from Core Facilities for Life and Environmental Sciences (State Key Laboratory of Microbial Technology, Shandong University). The cost of chiral derivatization prior to MS analysis is not included.

^bCalibrated based on the purchase expense of AlphaScreen donor and acceptor beads and the quoted price for sample testing from Core Facilities for Life and Environmental Sciences (State Key Laboratory of Microbial Technology, Shandong University). The cost of biotinylated DNA and DhdR is not included.

^cThe maximum number of samples in a single detection.

Minor comments:

General: Protein names used throughout when referring to genes, rather than italicised gene names. Extracellular and intracellular activity used when it was not clear if the authors mean exogenous and endogenous addition/production.

Response: Thanks for your good suggestion. We have changed protein names to italicised gene names when referring to genes. In addition, we have also changed “extracellular” and “intracellular” to “exogenous” and “endogenous” for clearer expression and easier understanding.

L27 “dehydrogenase” not “dehydrogenases”

Response: We have changed “dehydrogenases” to “dehydrogenase” in the revised manuscript.

L34 amplified luminescent proximity homogeneous assay technology (Alpha Screen).

Response: We have added “(AlphaScreen)” after “amplified luminescent proximity homogeneous assay” in the revised manuscript.

L49 would be useful to know the physiological relevant concentrations of D2HG upfront

Response: Thanks for your good suggestion. The physiological relevant concentrations of D-2-HG in healthy human plasma is < 0.9 μM according to references 1 and 10 in the revised manuscript. We have added the physiological relevant concentrations of D-2-HG in the revised manuscript as follows:

“D-2-HG is present at rather low levels (< **0.9 μM in the plasma of healthy humans**) under physiological conditions^{1,10}.”

In the section “References”:

1. Kranendijk, M., Struys, E.A., Salomons, G.S., Van der Knaap, M.S. & Jakobs, C. Progress in understanding 2-hydroxyglutaric acidurias. *J. Inherit. Metab. Dis.* **35**, 571–587 (2012).
10. Kranendijk, M. *et al.* Evidence for genetic heterogeneity in D-2-hydroxyglutaric aciduria. *Hum. Mutat.* **31**, 279–283 (2010).

L68 This aTF can repress the...

Response: We have changed “depress” to “repress” in the revised manuscript.

L69 ...and specifically is derepressed by D-2-HG.

Response: We have changed “respond to” to “is derepressed by” in the revised manuscript.

L75 what is D-2-HG generated from?

Response: D-2-HG is generated from 2-KG by UDP-2-acetamido-2-deoxy-D-glucuronic acid (UDP-GlcNAcA) 3-dehydrogenase, WbpB, of *Pseudomonas aeruginosa* PAO1. We have added “from 2-KG” after “the intracellular generation of D-2-HG” in the revised manuscript.

L82 Genes encoding GntR...

Response: We have changed the sentence “GntR family transcriptional regulator...” to “Genes encoding GntR family transcriptional regulators...” in the revised manuscript.

L88 GntR and CdaR - these are proteins, correct to "genes encoding GntR..." or change to italicised gene names

Response: We have changed “GntR” and “CdaR” to their corresponding italicised gene names “*gntR*” and “*cdaR*” in the revised manuscript.

L90, L91, L94, L95, L96 same as above - gene names required

Response: We have changed protein names in L90, L91, L94, L95, L96 of the previous manuscript to their corresponding italicised gene names according to your good suggestion.

L114 SerA appears to be a tetramer based on calibration curves ~ 160 kDa = 4 * 40kDa – may require sec-MALS to resolved accurate multimeric form

Response: Thanks for your reminding. We have corrected “dimer” to “tetramer” in the revised manuscript.

L116 enzyme kinetic parameters may need to be revised based on correct multimeric form

Response: We have revised the enzyme kinetic parameters of SerA based on the correct multimeric form and have changed the unit of V_{\max} to $\mu\text{M min}^{-1}$ as follows:

“It was also able to reduce 2-KG to D-2-HG (Supplementary Fig. 3c, d), and the apparent K_m , V_{\max} , and k_{cat} for 2-KG were $57.88 \pm 0.71 \mu\text{M}$, $4,062.02 \pm 41.74 \mu\text{M min}^{-1}$, and $5.79 \pm 0.06 \text{ s}^{-1}$, respectively (Supplementary Fig. 3e and Supplementary Table 2).”

Supplementary Figure 3. Purification and characterization of SerA in *A. denitrificans* NBRC 15125. e, The Lineweaver–Burk plot for the purified SerA toward 2-KG. The reaction mixtures contained 0.05 mg mL⁻¹ purified SerA, 0.2 mM NADH, and variable concentrations of 2-KG in 50 mM Tris-HCl (pH 7.4). The activity of SerA was assayed at 30 °C by measuring the oxidation of NADH spectrophotometrically at 340 nm. Data shown are mean \pm s.d. ($n = 3$ independent experiments).

Supplementary Table 2 Kinetic parameters of purified D2HGDH and SerA from *A. denitrificans* NBRC 15125^a.

Enzyme	Substrate	K_m (μM)	V_{\max} ($\mu\text{M min}^{-1}$)	k_{cat} (s^{-1})	k_{cat}/K_m ($\text{s}^{-1} \mu\text{M}^{-1}$)
D2HGDH ^b	D-2-HG	31.16 ± 1.41	$40,681.73 \pm 946.03$	6.90 ± 0.16	0.22 ± 0.00
SerA ^c	2-KG	57.88 ± 0.71	$4,062.02 \pm 41.74$	5.79 ± 0.06	0.10 ± 0.00

^aData shown are mean \pm s.d. ($n = 3$ independent experiments)

^bThe activity of D2HGDH was assayed at 30 °C by determining the reduction of DCPIP spectrophotometrically at 600 nm in 800 μL reaction mixture containing 50 mM Tris-HCl buffer (pH 7.4), 0.01 mg mL⁻¹ purified D2HGDH, 0.2 mM PMS, 0.05 mM DCPIP, and variable concentrations of D-2-HG.

^cThe activity of SerA was assayed at 30 °C by measuring the oxidation of NADH spectrophotometrically at

340 nm in 800 μ L reaction mixture containing 50 mM Tris-HCl buffer (pH 7.4), 0.05 mg mL⁻¹ purified SerA, 0.2 mM NADH, and variable concentrations of 2-KG.

L119 intracellular accumulation?

Response: The accumulation of D-2-HG in L119 refers to extracellular accumulation. We have added “extracellular” in the revised manuscript. We also measured the concentration of intracellular D-2-HG in *A. denitrificans* NBRC 15125 and its derivatives. The related data have been added in the revised manuscript as follows:

“No significant differences in intracellular D-2-HG concentrations were found between *A. denitrificans* NBRC 15125 and its derivatives (**Fig. 1e**).”

Figure 1 D2HGDH contributes to D-2-HG catabolism in *A. denitrificans* NBRC 15125. e, Growth, extracellular, and intracellular D-2-HG concentrations of *A. denitrificans* NBRC 15125 and its derivatives cultured in LB medium at mid-log stage. Data shown are mean \pm s.d. ($n = 3$ independent experiments).

L129 “D2HGDH is critical for extracellular D-2-HG utilization” is D2HGDH secreted? Not clear what is meant here relative to “lack intracellular metabolism” observed for D2HGDH see L122 – please clarify/rephrase this section

Response: We are sorry for our misleading description. As shown below, we predicted the localization of D2HGDH in *A. denitrificans* NBRC 15125 by SignalP 5.0 (DOI: 10.1038/s41587-019-0036-z). The prediction shows that the probability of D2HGDH has signal peptide is 3.44%, suggesting that D2HGDH is not secreted.

Prediction: Other

Protein type	Signal peptide (Sec/SPI)	TAT signal peptide (Tat/SPI)	Lipoprotein signal peptide (Sec/SPII)	Other
Likelihood	0.0183	0.0154	0.0007	0.9657

We have rephrased this section in the revised manuscript as follows:

“Disruption of *d2hgdh* abolished the ability of *A. denitrificans* NBRC 15125 ($\Delta d2hgdh$) to assimilate D-2-HG but did not affect growth in the presence of L-2-HG, D-malate, D-lactate, and 2-KG, indicating that D2HGDH is critical for the utilization of **exogenous** D-2-HG by this strain (Fig. 1g).”

L135 was RT-PCR perform in the absence of D2HG this would seem a logical experiment to perform to characterise native regulation/performance?

Response: Thanks for your good advice. We have performed RT-PCR using cDNA generated from the total RNA of cells grown in minimal medium containing 3 g L^{-1} D-2-HG or pyruvate as the sole carbon source (**Fig. 2a**) according to your suggestion. As shown in **Fig. 2a**, *dhdR* and *d2hgdh* genes were co-transcribed and the higher transcription of the fragment was detected when D-2-HG was used as the sole carbon source.

Figure 2 DhdR negatively regulates the catabolism of D-2-HG. **a**, Identification of the cotranscription of *dhdR* and *d2hgdh* in the presence D-2-HG or pyruvate by RT-PCR. Genomic DNA of *A. denitrificans* NBRC 15125 was used as a positive control (lanes C). RT-PCR was performed by using the cDNA as template (lanes +). RNA was used as negative control (lanes -). Lane M, DNA ladder marker. The amplicon locations are indicated in panel a.

L135-6 Where is the data to support the TSS?

Response: The related data to support the TSS has been added as **Supplementary Fig. 5b** in the revised manuscript and Supplementary Information as follows:

“The TSS was confirmed to be an adenine (A) residue that overlaps with the *dhdR* start codon, and the putative -10 (TATAAT) and -35 (TTATCA) regions are separated by 20 bp (Fig. 2b and **Supplementary Fig. 5**).”

Supplementary Figure 5. Determination of the transcriptional start site (TSS) of *dhdR* by rapid amplification of cDNA ends (RACE). **b**, Chromatogram displays the partial sequences of the 5' RACE

products, which are the complementary sequences of the nucleotide sequence shown in **a**. The TSS of *dhdR* is indicated by bent arrow.

L136 ...*dhdR* was identified by rapid amplification of cDNA ends (RACE).

Response: We have added “(RACE)” after “...*dhdR* was identified by rapid amplification of cDNA ends” in the revised manuscript.

L148 ... region overlaps with the...

Response: We have changed “overlapped” to “overlaps with” in the revised manuscript.

L166 “sensing” rather than “biorecognition” maybe a more appropriate term?

Response: We have changed “biorecognition” to “sensing” throughout the revised manuscript.

L183 remove “extra”

Response: The word has been deleted in the revised manuscript.

L199 add comment on the upper detection limit and dynamic range

Response: The upper detection limit and dynamic range have been added in the revised manuscript as follows:

“As shown in **Fig. 3d, e**, B_{D2HG-0} responded to D-2-HG supplementation in a dose-dependent manner, and the limit of detection (LOD) and linear range of B_{D2HG-0} were 0.50 μ M and 2–50 μ M, respectively (**Supplementary Table 4**).”

Figure 3 Design of the D-2-HG biosensor (B_{D2HG}). **d**, Response of B_{D2HG}-0 to different concentrations of D-2-HG. **e**, The linear detection range of B_{D2HG}-0. Black dotted line is a reference line where the reduced luminescence signal (Δ RLU) is three times of the background signal. The concentrations of DhdR and Bio-*dhdO* in **d–f** were 0.3 nM and 1 nM, respectively. Data shown are mean \pm s.d. ($n = 3$ independent experiments).

Supplementary Table 4 Key parameters of the developed biosensors for D-2-HG assay in HBS-P buffer.

Parameter	B _{D2HG} -0	B _{D2HG} -1	B _{D2HG} -2	B _{D2HG} -7	B _{D2HG} -8	B _{D2HG} -12
K_D (μ M) ^a	0.64	3.03	2.67	5.81	4.02	6.25
EC ₅₀ (μ M) ^b	9.05	1.33	2.69	1.19	1.62	1.68
LOD (μ M) ^c	0.50	0.10	0.50	1.00	1.00	1.00
Linear range (μ M)	2–50	0.3–20	0.5–10	0.5–10	1–10	1–20

^aThe equilibrium dissociation constant (K_D) was determined by ITC and analyzed by using a single-site binding model.

^bEC₅₀ indicates the concentration of D-2-HG producing a 50% signal reduction.

^cThe limit of detection (LOD) was the minimal concentration of D-2-HG where the reduced luminescence signal (Δ RLU) is at least 3 times of the background signal¹.

L215 for biosensor / effector terminology EC50 is more commonly used

Response: We have changed “I₅₀” to “EC50” in the revised manuscript.

L217 further expansion of the different outputs would have been interesting here. Lower affinity initially appears to increase output RLU signal i.e. designs 1 & 2 vs 0, but further decreasing in affinity leads a drop off in output - why? Further decreased affinity between the aTF and operator appears to result in more sensitive dose response curves. It would be useful here to understand the desired performance functions i.e. sensitivity, dynamic etc that you seeking to achieved. And therefore from design rules perspective it would be useful here to aid future

biomolecular engineers to how this may be achieved through DNA/promoter engineering etc

Response: Thanks for your good suggestion and we are sorry for our confusing description. Actually, decreasing in affinity leads to a drop off in output of all designs, including designs 1 & 2. The design rule and optimization strategy in this study was based on a mathematical modeling provided by **reference 25** in the revised manuscript. According to this mathematical modeling, reducing the affinity between the DhdR and its binding site (increasing equilibrium dissociation constant K_D) or increasing the affinity between DhdR and D-2-HG (reducing the equilibrium inhibition constant K_I) could lower I_{50} and increase k_{max} , which results in the improvement of sensitivity and broadening of the linear range. In this study, we chose point mutation of DBS which was relatively simple to increase K_D rather than complex protein engineering of DhdR to reduce K_I and obtained five optimized biosensors with different responses to D-2-HG. However, it is probably not most suitable for the construction of biosensors when the K_D value of DhdR for DBS is too high. The signal to noise ratio (S/N) of biosensor with the low affinity of DhdR for DBS would reduce and further influence the limit of detection (LOD). A short sentence was added in the revised manuscript as follows:

“However, it is probably not most suitable for the construction of biosensors when the K_D value of DhdR for DBS is too large. The S/N would reduce and influence the LOD of the biosensor due to the low affinity of DhdR for DBS.”

L230 ...in vivo, therefore these...

Response: Thanks for your good suggestion. We have rephrased this section in the manuscript as follows:

“2-KG, L-glutamate, L-glutamine, and isocitrate are possible precursors of D-2-HG *in vivo*. Therefore, these metabolites and L-2-HG (the enantiomer of D-2-HG) were also added to the assay system. As shown in Fig. 5e, f, 2-KG, L-glutamate, L-glutamine, isocitrate, and L-2-HG did not interfere with D-2-HG quantitation in serum or urine.”

L239 IDH1 - definition required what is this enzyme/gene - also use gene name

Response: IDH1 has been defined as isocitrate dehydrogenase 1 in the “Introduction” section. Here, “IDH1/R132H” has been changed to “*IDH1*/R132H” in the revised manuscript.

L240 gene name

Response: “IDH1/R132C” has been changed to “*IDH1*/R132C” in the revised manuscript.

L249 Based on – remove. ...The dose-response curves...

Response: The words have been deleted in the revised manuscript.

L251 “assayed” replace with “determined”

Response: We have replaced “assayed” with “determined” in the revised manuscript.

L333 does D2HGDH have a signal peptide? Clarify what extracellular catabolism means

Response: D2HGDH does not have a signal peptide according to the prediction of SignalP 5.0 as mentioned above. The “extracellular D-2-HG catabolism” in the previous manuscript means utilization of exogenous added D-2-HG or utilization of D-2-HG existed in external environment. We have changed “extracellular” to “exogenous” in the revised manuscript.

L495, L499, L517 primer sequence and amplicon locations required

Response: Thanks for your suggestion. All the sequence of primers used in this study have been listed in the **Supplementary Data 2** for clearly and easily search, including the primer sequences of L495, L499, L517 (in the previous manuscript). We have added amplicon locations as follows:

For L495 of the previous manuscript: “The amplicon was designed to be a 301-bp fragment and located overlapping with *d2hgdh* and *dhdR* genes.”

For L499 of the previous manuscript:

a

```

CGTCGAACCCGAGGACAAAATGCTGCCGGACGGCGAGCGCGCCGTGGCCT
GAGTCGCGGGCGGCGCGCCGGATCCGGGCTGTCATTGTCAAAAGTTATCAGA
TAACCTGAAAAGTAGGCCCTATAAATCGGGCGC+1ATGCTGAGCAAGAGCCTGACC-35
-10
TTGACCGAACAGGTCGCCCCGCCAGATCGCGGGCGACATCGCCGAAGGCGT
CCATTCCGTGGGCGCCAAGCTGCCGCCCGGCGGTGTCTTGGCGGAGCAGT
ACGGTGTGAGCGCCGCGGTTCATCCGCGAGGCCACCGAGCGCCTGCGCGCC
CAGGGGCTGATCCAGAGCCGCCAGGGCTCGGGCAGCGTGGTGGTGTCCC
CACCGTGCTCAGGGCTTCCAGGTTTCCGCCGGCCTCGACGATCGCGAGC
AGCTGGCCAGCGTCTACGAATTGCGGATGGAAGGCGGGCGCGGCC
GCCCTGGCGGGGAGGCGCCGCAACGCCACCGACCTTGGCGCCATGGCCGA
GGCCCTGGCCGCGCTGGAAGCGAACCTGGACCATCGGAACAGGGCGTC
GAGCACGACATCGCCTTCCACGTCGCCATCGCCGCCGCCACGCACAACCGT
TATTACCAGGACCTGCTGCAGTACCTGAACCTGCAGCTGCGCCTGGCCGTC
AGCACCGCGCGCACCAACAGCCGCCGTCAGGAGGGCCTGACCGCGGTGGT
GCACCAGGAACACGTGGCCGTCTACGACGCCATCCTCGCGGGCGATCCCG
ACCGCGCCCCACTGGCGGCGACCCGCCACTTGCAGCAGGCGGCCAGCCG
CCTGCGTCTCGATCTCCTCTCCTCCGGCCGCAAGGCAGACATCATGA

```

Supplementary Figure 5. Determination of the transcriptional start site (TSS) of *dhdR* by rapid amplification of cDNA ends (RACE). a, Map of the intergenic region upstream of *dhdR*-*d2hgdh* operon and *dhdR*. TSS is shown in enlarged red letter and marked by +1. The putative -35 and -10 regions are shown in bold and underlined. Amplicon locations are indicated by primers GSP1, GSP2, and GSP3.

For L517 of the previous manuscript: “The FAM-labeled probe was amplified with primers M13F-FAM (priming site: bases 290–305 of pEASY-Blunt Simple Cloning Vector) and M13R (priming site: bases 205–221 of pEASY-Blunt Simple Cloning Vector) and plasmid pEASY-Blunt-F1 as template.”

L972 cell line descriptions required here

Response: We have added cell line descriptions in the revised manuscript as follows:

“i, D-2-HG detection in cell culture medium of HEK293FT cells, HEK293FT-IDH1/R132H cells, and HT1080 cells upon treatment with 0.5 μM GSK 864 or AGI-6780, respectively.”

Fig1d – chemical structures would be useful – see below (Figure 3a)

Response: We have added the chemical structures in **Fig. 1d** and **Fig. 2i** (**Fig. 3a** in the

previous manuscript) in the revised manuscript.

Figure 1 D2HGDH contributes to D-2-HG catabolism in *A. denitrificans* NBRC 15125. **d**, HPLC analysis of the product of D2HGDH-catalyzed D-2-HG dehydrogenation. The reaction mixtures containing D-2-HG (1 mM), 3-(4,5-dimethylthiazol-2-yl)-2,5-diphenyltetrazolium bromide (MTT, 5 mM) and active or denatured D2HGDH (1 mg mL⁻¹) in 50 mM Tris-HCl (pH 7.4) were incubated at 37 °C for 5 min. Black line, the reaction with denatured D2HGDH; red line, the reaction with active D2HGDH.

Figure 2 DhdR negatively regulates the catabolism of D-2-HG. **i**, Proposed model for the regulation of D-2-HG catabolism by DhdR in *A. denitrificans* NBRC 15125. DhdR represses the expression of *dhdR-d2hgdh* genes. D-2-HG is the effector of DhdR and prevents DhdR binding to the *dhdR* promoter region.

Fig 1e – re-scaled concentration y-axis

Response: We have added the concentration of intracellular D-2-HG of *A. denitrificans* NBRC 15125 and its derivatives. Accordingly, the concentration y-axis has been re-scaled.

Figure 1 D2HGDH contributes to D-2-HG catabolism in *A. denitrificans* NBRC 15125. e, Growth, extracellular, and intracellular D-2-HG concentrations of *A. denitrificans* NBRC 15125 and its derivatives cultured in LB medium at mid-log stage. Data shown are mean \pm s.d. ($n = 3$ independent experiments).

Fig 2a - how is this demonstrated? amplicon location details required - please add

Response: This is demonstrated by RT-PCR with primers listed in **Supplementary Data 2**. cDNA generated from the total RNA of cells grown in minimal medium containing 3 g L^{-1} D-2-HG or pyruvate as the sole carbon source was used as template. Genomic DNA of *A. denitrificans* NBRC 15125 was used as a positive control. RNA extracted from cells grown in minimal medium containing 3 g L^{-1} D-2-HG or pyruvate was used as negative control. The fragment amplified was a 301-bp fragment overlapped with *dhdR* and *d2hgdh* genes. The amplicon location details have been added in the revised manuscript as follows:

“To determine whether *dhdR* and *d2hgdh* form an operon, a 301-bp fragment overlapped with *dhdR* and *d2hgdh* was amplified by reverse transcription-PCR (RT-PCR) using cDNA generated from the total RNA of cells grown in minimal medium containing 3 g L^{-1} D-2-HG or pyruvate as the sole carbon source. As shown in Fig. 2a, the intergenic *dhdR*-*d2hgdh* were successfully amplified, suggesting that these two genes are co-transcribed.”

Fig3a - figure panel refers to the regulation of operon - would seem to fit better with the previous figure / section rather than the 'biosensor' work – also include chemical structures.

Response: We have moved the figure panel (**Fig. 3a** in the previous manuscript) into **Fig. 2** and added the chemical structure in this figure panel (**Fig. 2i** in the revised manuscript) according to your good suggestion.

Figure 2 DhdR negatively regulates the catabolism of D-2-HG. i, Proposed model for the regulation of D-2-HG catabolism by DhdR in *A. denitrificans* NBRC 15125. DhdR represses the expression of *dhdR-d2hgdh* genes. D-2-HG is the effector of DhdR and prevents DhdR binding to the *dhdR* promoter region.

Supp Figure 9-1 “D1” the isotherm does not seem to correspond to the displayed fitted data below - raw data indicates no binding? check and provide more convincing data

Response: Thanks for your good question. As shown in **Supplementary Fig. 13b**, the titration of D1 into DhdR was an exothermic reaction while the titration of D1 into buffer (control experiment) was an endothermic reaction. The net heat of the dilutions was corrected by subtracting the heat of the control point-to-point and was corrected prior to curve fitting, therefore the experimental isotherm does not seem to correspond to the displayed fitted data in the panel of curve fitting. We have provided the isotherm of control experiment of ITC and corrected isotherm as more convincing data in the revised manuscript.

Supplementary Figure 13. Determination of affinities of DhdR with 27-bp DNA fragments containing DBS or DBS mutants by ITC. b, Titration of D1 into DhdR. 100 μM DNA fragment was titrated into 15 μM DhdR with 19 injections. The control experiment was performed by titrating the corresponding DNA fragment into reaction buffer. The net heat of the dilutions was corrected by subtracting the heat of the control point-to-point. A one site fitting model was used for curve fitting by using MicroCal PEAQ-ITC analysis software.

Reviewer #2 (Remarks to the Author):

This study from Xiao et al describes the development and application of a novel D-2-HG biosensor based on a bacterial allosteric transcription factor which they name DhdR. They leverage the AlphaScreen approach to create a bead-based assay with high selectivity for D-2-HG and use this to probe in biologically relevant conditions. Further they apply the biosensor to identify a potential role for PAOI in Pseudomonas aeruginosa as a protein that participates in D-2-HG synthesis. I have only extremely minor comments with respect to the work.

Response: Thanks for your kind comments.

(1) 13 point mutants are generated in order to better optimize the biosensor, yielding a significant decrease in LOD to 0.08uM. Is there a rationale for why this point mutation was efficacious?

Response: The optimization strategy for our biosensors was based on a mathematical modeling provided by **reference 25** in the revised manuscript. According to this mathematical modeling, reducing the affinity between the aTF and TFBS (increasing K_D) could lower I_{50} and increase k_{max} , which contributed to the improvement of the sensitivity and the broaden of the linear range of biosensors. Thus, point mutation was adopted to reduce the affinity between DhdR and its binding site to optimize the limit of detection (LOD) of our biosensor. We have added the citation of this reference as follows:

“Reducing the affinity between DhdR and its binding site could increase the equilibrium dissociation constant K_D and finally improve the sensitivity and linear detection range of the biosensor²⁵.”

In the section “References”:

25. Li, S. *et al.* A platform for the development of novel biosensors by configuring allosteric transcription factor recognition with amplified luminescent proximity homogeneous assays. *Chem. Commun.* **53**, 99–102 (2017).

(2) The authors demonstrate the biosensor’s sensitivity in the context of L-2-HG generation in IDH1 mutant HEK293T cells, though it is unclear how much 2-HG (L v D) is generated in

these cells. It is known that the relative levels of L-2-HG can be quite high relative to D-2-HG, sometimes more than 1000-fold with L-2-HG being in the mM. Have the authors tested the ability of the sensor to accurately quantify μM concentrations of D-2-HG in this context?

Response: Thanks for your good suggestion. We have tested the ability of B_{D2HG}-1 to accurately quantify μM concentrations of D-2-HG by adding 1 mM L-2-HG into cell culture medium and then determining the dose-response curve of B_{D2HG}-1 in 10-fold diluted cell culture medium. As shown in **Supplementary Fig. 15**, addition of 1 mM L-2-HG into cell culture medium could barely affect the response of B_{D2HG}-1 to D-2-HG. This data has been added in the revised manuscript.

Supplementary Figure 15. Effects of L-2-HG in cell culture medium on the quantification of D-2-HG.

The concentration of L-2-HG in cell culture medium was 1 mM. Dose-response curves of B_{D2HG}-1 were determined in 10-fold diluted cell culture medium in the presence (red line) or absence (black line) of L-2-HG. Data shown are mean \pm s.d. ($n = 3$ independent experiments).

(3) Can the authors comment on the ability to use such an approach as an intracellular biosensor coupled to other technologies for real-time imaging of D-2-HG production?

Response: Thanks for your good question. Because streptavidin donor and nickel chelate acceptor beads can not penetrate the cell membrane, the approach developed in this study could not be used as an intracellular biosensor for real-time imaging of D-2-HG production. However, FRET sensors and single fluorescent protein sensors with specific regulators have been widely

used for the real-time monitoring of the intracellular levels of various metabolites, such as lactate (DOI: 10.1371/journal.pone.0057712), pyruvate (DOI: 10.1371/journal.pone.0085780), formaldehyde (DOI: 10.1038/s41467-020-20754-4), and hydrogen peroxide (DOI: 10.1016/j.cmet.2020.02.003). DhR is the first reported transcription regulator specifically responding to D-2-HG across all domains of life. Therefore, FRET sensors or single fluorescent protein-based sensors using DhR as the sensing element may be developed and used in real-time imaging of intracellular D-2-HG production.

A short sentence has been added in “Discussion” section of the revised manuscript as follows:

“FRET sensors and single fluorescent protein sensors with specific regulators have been widely used for the real-time monitoring of the intracellular levels of various metabolites⁴⁴⁻⁴⁷. D-2-HG-sensing fluorescent reporters using DhR as the sensing element may be developed and used in real-time imaging of intracellular D-2-HG production.”

In the section “References”:

44. San Martín, A. *et al.* A genetically encoded FRET lactate sensor and its use to detect the Warburg effect in single cancer cells. *PLoS One* **8**, e57712 (2013).
45. San Martín, A. *et al.* Imaging mitochondrial flux in single cells with a FRET sensor for pyruvate. *PLoS One* **9**, e85780 (2014).
46. Zhu, R. *et al.* Genetically encoded formaldehyde sensors inspired by a protein intrahelical crosslinking reaction. *Nat. Commun.* **12**, 581 (2021).
47. Pak, V.V. *et al.* Ultrasensitive genetically encoded indicator for hydrogen peroxide identifies roles for the oxidant in cell migration and mitochondrial function. *Cell Metab.* **31**, 642–653.e6 (2020).

Reviewer #3 (Remarks to the Author):

This study describes the identification of a transcriptional regulator that recognizes D-2-HG and the development and analysis of a series of derived biosensor. This is an interesting study with significant potential. However, this reviewer has detected several major and minor flaws and inconsistencies, listed below, that have to be addressed

Response: Thanks for your good comments and suggestions. We have made corresponding revisions according to your suggestions.

In the introduction the authors state that “However, chiral derivatization with proper reagents is necessary to distinguish D-2-HG from its mirror-image enantiomer L-2-HG, and these methods are time-consuming and laborious.” It would be fair to add that both isomers can be probably resolved by ion-mobility qTOF mass spectrometry, without any derivatization.

Response: Thanks for your kind advice. We have rephrased the sentence as follows:

“Gas chromatography-tandem mass spectrometry (GC-MS/MS)^{18,19} and liquid chromatography-tandem mass spectrometry (LC-MS/MS)^{20,21} are often used to quantitatively assess D-2-HG levels. Chiral derivatization with proper reagents is necessary to distinguish D-2-HG from its mirror-image enantiomer L-2-HG by GC-MS/MS and LC-MS/MS²². Ion mobility-quadrupole-time-of-flight mass spectrometry (IM-QTOF-MS), a powerful tool to separate complex mixtures, may resolve both enantiomers without any derivatization²³.”

In the section “References”:

22. Yuan, B.-F. Quantitative analysis of oncometabolite 2-hydroxyglutarate. *Adv. Exp. Med. Biol.* **1280**, 161–172 (2021).
23. Lanucara, F., Holman, S.W., Gray, C.J. & Eyers, C.E. The power of ion mobility-mass spectrometry for structural characterization and the study of conformational dynamics. *Nat. Chem.* **6**, 281–294 (2014).

The authors determined V_{max} of the D-2-HGDH enzyme and provide U / mg as unit. However, $V_{max} = K_{cat} \times E(t)$ (total enzyme concentration), therefore the unit for V_{max} is typically $\mu\text{M}/\text{min}$. The unit U/mg is the unit for the specific enzyme activity. V_{max} and

specific enzyme activity are not the same. This needs to be corrected.

Response: Thanks for your good suggestion. We have recalculated V_{\max} of the D2HGDH and SerA, and corrected the unit of V_{\max} to microM/min as follows:

“The apparent K_m , V_{\max} , and k_{cat} of purified D2HGDH for D-2-HG were $31.16 \pm 1.41 \mu\text{M}$, $40,681.73 \pm 946.03 \mu\text{M min}^{-1}$, and $6.90 \pm 0.16 \text{ s}^{-1}$, respectively (**Supplementary Fig. 2** and **Supplementary Table 2**)”

“It was also able to reduce 2-KG to D-2-HG (Supplementary Fig. 3c, d), and the apparent K_m , V_{\max} , and k_{cat} for 2-KG were $57.88 \pm 0.71 \mu\text{M}$, $4,062.02 \pm 41.74 \mu\text{M min}^{-1}$, and $5.79 \pm 0.06 \text{ s}^{-1}$, respectively (**Supplementary Fig. 3e** and **Supplementary Table 2**).”

Supplementary Figure 2. Lineweaver–Burk plot for the purified D2HGDH toward D-2-HG. The reaction mixtures contained 0.01 mg mL^{-1} purified D2HGDH, 0.2 mM PMS, 0.05 mM DCPIP, and variable concentrations of D-2-HG in 50 mM Tris-HCl (pH 7.4). The activity of D2HGDH was assayed at $30 \text{ }^\circ\text{C}$ by determining the reduction of DCPIP spectrophotometrically at 600 nm . Data shown are mean \pm s.d. ($n = 3$ independent experiments).

Supplementary Figure 3. Purification and characterization of SerA in *A. denitrificans* NBRC 15125. **e**, The Lineweaver–Burk plot for the purified SerA toward 2-KG. The reaction mixtures contained 0.05 mg mL⁻¹ purified SerA, 0.2 mM NADH, and variable concentrations of 2-KG in 50 mM Tris-HCl (pH 7.4). The activity of SerA was assayed at 30 °C by measuring the oxidation of NADH spectrophotometrically at 340 nm. Data shown are mean ± s.d. ($n = 3$ independent experiments).

Supplementary Table 2 Kinetic parameters of purified D2HGDH and SerA from *A. denitrificans* NBRC 15125^a.

Enzyme	Substrate	K_m (μM)	V_{max} (μM min ⁻¹)	k_{cat} (s ⁻¹)	k_{cat}/K_m (s ⁻¹ μM ⁻¹)
D2HGDH ^b	D-2-HG	31.16 ± 1.41	40,681.73 ± 946.03	6.90 ± 0.16	0.22 ± 0.00
SerA ^c	2-KG	57.88 ± 0.71	4,062.02 ± 41.74	5.79 ± 0.06	0.10 ± 0.00

^aData shown are mean ± s.d. ($n = 3$ independent experiments).

^bThe activity of D2HGDH was assayed at 30 °C by determining the reduction of DCPIP spectrophotometrically at 600 nm in 800 μL reaction mixture containing 50 mM Tris-HCl buffer (pH 7.4), 0.01 mg mL⁻¹ purified D2HGDH, 0.2 mM PMS, 0.05 mM DCPIP, and variable concentrations of D-2-HG.

^cThe activity of SerA was assayed at 30 °C by measuring the oxidation of NADH spectrophotometrically at 340 nm in 800 μL reaction mixture containing 50 mM Tris-HCl buffer (pH 7.4), 0.05 mg mL⁻¹ purified SerA, 0.2 mM NADH, and variable concentrations of 2-KG.

Electrophoretic mobility shift: Although the EMSA data are clear, there were no controls included. Typically, EMSA controls involve the binding to non-specific DNA or competition

assays with the specific, unlabeled DNA fragment.

Response: Thanks for your good suggestion. According to your advice, we have performed EMSA with an internal fragment of *dhdR* as non-specific DNA control. As shown in **Supplementary Fig. 7**, the 81-bp fragment (F1) upstream of *dhdR* formed DhdR-F1 complexes with purified DhdR, while the internal fragment of *dhdR* could not interact with purified DhdR.

Supplementary Figure 7. EMSAs with 81-bp fragment (F1) upstream of *dhdR* (10 nM) and purified DhdR (0, 10, 20, 30, 40, 50, 60, 80, and 100 nM). A 200-bp internal fragment of *dhdR* (10 nM) was used as a negative control. The position of free F1, complex of DhdR-F1 (C), and negative control (N) are indicated. Lane M, DNA ladder marker.

The authors state that D-2-HG is the specific effector of DhdR. However, this statement is based solely on the analysis of 6 other compounds. To consolidate this claim further experiments are required. This referee recommends the use of thermal shift assays of the transcriptional regulator in the presence of compounds from compound libraries such as Biolog compound arrays. Such assays can be conducted in a high-throughput format using 96 well plates.

Response: Thanks for your good suggestion. We have conducted thermal shift assays with 30 compounds to consolidate the claim that D-2-HG is the specific effector of DhdR (**Supplementary Fig. 8**). These 30 compounds are consisting of D-2-HG analogues, metabolites involved in the tricarboxylic acid cycle, metabolites within D-2-HG metabolic pathways and several saccharides. As shown in **Supplementary Fig. 8**, only D-2-HG lead to a significant increase of T_m relative to that of DhdR without ligands ($\Delta T_m = 7.5$ °C), suggesting

that D-2-HG is the specific effector of DhdR.

Supplementary Figure 8. Identification of the ligand of DhdR by FTS assays. The ΔT_m values indicate the changes in T_m values compared to control (DhdR without ligand). The concentration of tested compounds was 100 μM . Data shown are mean \pm standard deviations (s.d.) ($n = 3$ independent experiments).

Isothermal titration calorimetry was used to determine the thermodynamic binding parameters of DhdR. This manuscript would benefit from ITC studies of the binding of D2-HG to DhdR.

Response: Thanks for your good suggestion. The ITC studies of the binding of D-2-HG have been included in the revised manuscript (**Supplementary Fig. 9**) as follows:

“The result of isothermal titration calorimetry (ITC) also indicates that D-2-HG binds to DhdR ($K_D = 1.16 \pm 0.16 \mu\text{M}$) (**Supplementary Fig. 9**).”

Supplementary Figure 9. Isothermal titration calorimetry (ITC) of D-2-HG binding to Dh dR. **a**, Experiment isotherm. 500 μM D-2-HG was titrated into 40 μM Dh dR with 19 injections. **b**, Control isotherm. 500 μM D-2-HG was titrated into reaction buffer. **c**, Corrected isotherm. The net heat of dilutions was corrected by subtracting the heat of the control point-to-point. **d**, Curve fitting. A one site fitting model was used for curve fitting by using MicroCal PEAQ-ITC analysis software.

From Fig. 3A an I_{50} value (half maximal signal) of approximately 10 μM D-2-HG was determined. These data appear to contrast with Fig. 3C showing that 10 μM D2-HG caused only minor changes.

Response: Thanks for your kind suggestion and we are sorry for our misleading description. The I_{50} value of 9.05 μM (approximately 10 μM) D-2-HG shown in **Fig. 3d** in the revised manuscript was determined by using 0.3 nM Dh dR and 1 nM Bio-*dhdO* after optimization through cross-titration (**Fig. 3c** in the revised manuscript). The minor changes caused by 10 μM D-2-HG in **Fig. 3b** in the revised manuscript was determined with 1 nM Dh dR and 1 nM Bio-*dhdO*. The concentration of Dh dR used in **Fig. 3b** was almost 3.3 times of the concentration of Dh dR used in **Fig. 3d**.

A short sentence was added in the legend of **Fig. 3** as follows:

“The concentrations of DhdR and Bio-*dhdO* in **d–f** were 0.3 nM and 1 nM, respectively.”

Figure 3 Design of the D-2-HG biosensor (B_{D2HG}). **b**, Evaluation of allosteric effect of D-2-HG on *dhdO*-DhdR interaction by AlphaScreen. Data shown are mean ± s.d. ($n = 3$ independent experiments). **c**, Determination of the optimal concentrations of DhdR and Bio-*dhdO* by cross-titration. Each heat map cell shows the average of the luminescence signals of each titration ($n = 3$ independent experiments). **d**, Response of B_{D2HG-0} to different concentrations of D-2-HG. The concentrations of DhdR and Bio-*dhdO* in **d–f** were 0.3 nM and 1 nM, respectively. Data shown are mean ± s.d. ($n = 3$ independent experiments).

A key parameter of any analytical technique is the sensitivity or onset of response, i.e. the lowest concentration at which the presence of a given compound can be detected with confidence. The authors claim that this value is of 0.8 microM for the initial biosensor. However, considering that the I50 is about 10 microM and considering the data shown in Fig. 3e, it is unrealistic to state that the sensitivity of the biosensor is at 0.8 microM.

Response: Thanks for your professional comment. The limit of detection (LOD) in the previous version of the manuscript was calculated by interpolating the RLU_{LOD} into the dose-response curve. The RLU_{LOD} was calculated based on the following formula:

$$RLU_{LOD} = \text{average } R_{\max} - 3 \times \text{standard deviation}$$

where R_{\max} refers RLU in the absence of D-2-HG. This calculation method was provided

by the technical engineer of PerkinElmer. The LOD of B_{D2HG-0} was calculated to be 0.8 μM according to this calculation method.

The D-2-HG biosensor is developed according to the strategy in **reference 25** in the revised manuscript. Allosteric transcription factors HucR has been combined with AlphaScreen technology for the development of uric acid biosensors in **reference 25**, and the LOD has been defined as follows:

site fit K_I and one-site fit LogI_{50} models of software GraphPad Prism (version 5.0), respectively. Limit of detection (LOD) of biosensor was the minimal concentration of UA where the declined luminescence signal (ΔRLU) is at least three times of the blank.⁵

Thus, we have redefined LOD according to the **reference 25** as the minimal concentration of D-2-HG where the reduced luminescence signal (ΔRLU) is at least 3 times of the background signal in the revised manuscript.

Figure 3 Design of the D-2-HG biosensor (B_{D2HG}). e, The linear detection range of B_{D2HG-0}. Black dotted line is a reference line where the reduced luminescence signal (ΔRLU) is three times of the background signal. The concentrations of DhR and Bio-*dhdO* in **d–f** were 0.3 nM and 1 nM, respectively. Data shown are mean \pm s.d. ($n = 3$ independent experiments).

Reference:

25. Li, S. *et al.* A platform for the development of novel biosensors by configuring allosteric transcription factor recognition with amplified luminescent proximity homogeneous assays. *Chem. Commun.* **53**, 99–102 (2017).

The biosensor was then improved by mutation the operator site. The dose response for the best

biosensor is shown in Fig. 4b and the authors claim that the sensitivity of the system is of 0.08 microM. However, inspection of Fig. 4b shows that at 0.08 microM is in the noise area, particularly taking into account the error associated with the data at 1 microM. In addition, Supp. 10b (of the same optimized biosensor) reveals that the data obtained for 0.1 and 0.5 are basically the same and close to zero (background signal), a fact that is incompatible with the claim that the biosensor has a sensitivity of 0.08 microM.

Response: Thanks for your professional comment and we are very sorry for our misleading illustration. The LOD of B_{D2HG-1} (the best biosensor is shown in **Fig. 4b**) was previously calculated according to the method provided by the technical engineer from PerkinElmer. The Δ RLU at 0.1 μ M and 0.5 μ M were 2,661 and 11,103, respectively, which are higher than background signal (886 RLU).

We have re-determined the dose-response curve and linear detection range of B_{D2HG-1} and revised the LOD to 0.1 μ M according to the definition of LOD in **reference 25** in the revised manuscript. In addition, we have added a reference line which indicated 3 times of the background signal in **Fig. 4g** in the revised manuscript as follows:

Figure 4 Optimization of the D-2-HG biosensor (B_{D2HG}). **b–f**, Responses of the optimized biosensors to different concentrations of D-2-HG. **b**, Dose-response curve of B_{D2HG-1}; **g**, The linear detection range of B_{D2HG-1}. Black dotted line is a reference line where the reduced luminescence signal (Δ RLU) is three times of the background signal. The concentrations of DhDR and Bio-*dhdO* mutants in **b–g** were 0.3 nM and 1 nM, respectively. Data shown are mean \pm s.d. ($n = 3$ independent experiments).

Supplementary Table 4 Key parameters of the developed biosensors for D-2-HG assay in HBS-P buffer.

Parameter	B _{D2HG-0}	B _{D2HG-1}	B _{D2HG-2}	B _{D2HG-7}	B _{D2HG-8}	B _{D2HG-12}
K_D (μM) ^a	0.64	3.03	2.67	5.81	4.02	6.25
EC_{50} (μM) ^b	9.05	1.33	2.69	1.19	1.62	1.68
LOD (μM) ^c	0.50	0.10	0.50	1.00	1.00	1.00
Linear range (μM)	2–50	0.3–20	0.5–10	0.5–10	1–10	1–20

^aThe equilibrium dissociation constant (K_D) was determined by ITC and analyzed by using a single-site binding model.

^b EC_{50} indicates the concentration of D-2-HG producing a 50% signal reduction.

^cThe limit of detection (LOD) was the minimal concentration of D-2-HG where the reduced luminescence signal (ΔRLU) is at least 3 times of the background signal¹.

In the section “Supplementary Reference”:

1. Li, S. *et al.* A platform for the development of novel biosensors by configuring allosteric transcription factor recognition with amplified luminescent proximity homogeneous assays. *Chem. Commun.* **53**, 99–102 (2017).

There is a major flaw in ITC data interpretation. Data are not valid since not corrected for dilution heats prior to curve fitting. In some cases dilution heats are very important.

Response: Thanks for your good suggestion. In the previous version of the manuscript, dilution heats (except for D7, D9, D10, and D11) were indeed corrected before curve fitting. The control experiments were performed by titrating the corresponding DNA fragments into reaction buffer to quantify dilution heats. As for the data for D7, D9, D10, and D11, MicroCal PEAQ-ITC analysis software judged that there was no binding between these mutant DNA and DhdR. Thus, the dilution heats of D7, D9, D10, and D11 were not corrected in the previous version of the manuscript. According to your kindly suggestion, we also corrected the dilution heats of D7, D9, D10, and D11. In addition, **we provided raw data of ITC experiment, raw data of control experiment, data with corrected dilution heats and fitted curve for every mutant DNA as follows:**

Supplementary Figure 13-1. Determination of affinities of Dh4R with 27-bp DNA fragments containing DBS or DBS mutants by ITC. a, Titration of D0 into Dh4R. **b,** Titration of D1 into Dh4R. **c,** Titration of D2 into Dh4R. **d,** Titration of D3 into Dh4R. **e,** Titration of D4 into Dh4R. 100 μM DNA fragment was titrated into 15 μM Dh4R with 19 injections. The control experiment was performed by titrating the corresponding DNA fragment into reaction buffer. The net heat of the dilutions was corrected by subtracting the heat of the control point-to-point. A one site fitting model was used for curve fitting by using MicroCal PEAQ-ITC analysis software.

Supplementary Figure 13-2. Determination of affinities of Dh dR with 27-bp DNA fragments containing DBS or DBS mutants by ITC. **f**, Titration of D5 into Dh dR. **g**, Titration of D6 into Dh dR. **h**, Titration of D7 into Dh dR. **i**, Titration of D8 into Dh dR. **j**, Titration of D9 into Dh dR. 100 μM DNA fragment was titrated into 15 μM Dh dR with 19 injections. The control experiment was performed by titrating the corresponding DNA fragment into reaction buffer. The net heat of the dilutions was corrected by subtracting the heat of the control point-to-point. A one site fitting model was used for curve fitting by using MicroCal PEAQ-ITC analysis software.

Supplementary Figure 13-3. Determination of affinities of DhdR with 27-bp DNA fragments containing DBS or DBS mutants by ITC. **k**, Titration of D10 into DhdR. **l**, Titration of D11 into DhdR. **m**, Titration of D12 into DhdR. **n**, Titration of D13 into DhdR. 100 μM DNA fragment was titrated into 15 μM DhdR with 19 injections. The control experiment was performed by titrating the corresponding DNA fragment into reaction buffer. The net heat of the dilutions was corrected by subtracting the heat of the control point-to-point. A one site fitting model was used for curve fitting by using MicroCal PEAQ-ITC analysis software. No apparent interaction was detected between DhdR and D10.

Figure 4 Optimization of the D-2-HG biosensor (B_{D2HG}). a, Influence of point mutations in DBS on affinity determined by ITC. The point mutation is shown in red letter. The 1-bp interval in palindrome sequence is shown in gray letter. n.d., not detected.

At line 223 the authors state that the concentration of D2-HG has the potential to be a clinical indicator. It would be of interest to know this concentration in human samples and relate it to the sensitivity of the biosensor. To assess the performance of the biosensor the authors have spiked human serum and urine with D-2-HG and quantified the compound by ms and the developed biosensor. As detailed in the introduction the main motivation of this work is the development of a biosensor of an oncometabolite. The main concern of this reviewer is the lacking proof of concept of this biosensor to detect this oncometabolite. The authors have spiked human serum with D-2-HG and have detected this metabolite in the corresponding samples. However, this is not a proof of concept of this biosensor can be used to detect oncometabolite. To this issue samples from cancer patients and samples from healthy individuals need to be analysed to assess whether the biosensor is indeed able to differentiate these samples.

Response: Thanks for your good suggestions. In this study, we discovered the first transcription regulator responding to D-2-HG, developed a biosensor for D-2-HG detection, and identified the role of D-2-HG metabolism in lipopolysaccharide biosynthesis of *Pseudomonas aeruginosa*. To assess the performance of the biosensor in D-2-HG detection, we used the biosensor to assay the concentrations of D-2-HG in body fluids, different cell lines and bacterial minimal medium. Just as you mentioned above, the body fluids from cancer patients with IDH

mutations are the most ideal samples for the confirmation of clinical application of biosensor. However, we hope you will understand our inability to get the biological samples from patients with D-2-HG-related cancers, in spite of the continuous efforts we have made in the last six months.

Human serum spiked with target determinands is a commonly used alternative for the simulation of clinical samples. For example, commercial human serum (Beijing Solarbio Science & Technology Co., Ltd.) was spiked with uric acid and used for the evaluation of the performance of uric acid biosensors constructed based on the transcriptional regulator HucR in clinical samples (DOI: 10.1126/sciadv.aau4602). Therefore, we also used commercial human serum (purchased from Beijing Solarbio Science & Technology Co., Ltd.) and urine from the experiment operator spiked with D-2-HG to simulate samples from patients with D-2-HG-related disease. The physiological concentrations of D-2-HG in healthy human plasma is $< 0.9 \mu\text{M}$ and the concentrations of D-2-HG in serum of acute myeloid leukemia (AML) patients with IDH1-R132H and IDH2-R140Q are $63.6\text{--}870.9 \mu\text{M}$ and $54.9 \mu\text{M}$, respectively (DOI: 10.1111/j.1600-0609.2010.01505.x). We determined the concentration of D-2-HG in the biological sample spiked with D-2-HG using B_{D2HG}-1 and LC-MS/MS. The minimum concentration of D-2-HG in these samples was set to be $50 \mu\text{M}$, which is slight lower than the lower limit of the concentration of D-2-HG in cancer patients. As shown in **Fig. 5c, d**, B_{D2HG}-1 is able to detect the concentrations of D-2-HG in these simulated samples and the results of B_{D2HG}-1 showed high consistence with the results of LC-MS/MS (**Fig. 5c, d**).

The related discussion in the revised manuscript was as follows:

“Human serum spiked with target determinands is a commonly used alternative for the simulation of clinical samples. For example, commercial human serum spiked with uric acid was used for the evaluation of the performance of uric acid biosensors constructed based on the transcriptional regulator HucR in clinical samples²⁶. In this study, the concentration of D-2-HG spiked into serum and urine, and the amount of D-2-HG accumulated in different cell lines could be readily determined using B_{D2HG}-1. The results were highly concordant with those of LC-MS/MS.”

Figure 5 Measuring D-2-HG concentrations in various biological samples by using B_{D2HG-1}. **c,** Comparisons of analysis of D-2-HG in human serum by LC-MS/MS and B_{D2HG-1}. **d,** Comparisons of analysis of D-2-HG in urine by LC-MS/MS and B_{D2HG-1}. Black dotted line indicates a reference line.

In addition, a revision of the English is required, in particular the use of articles and the use of tenses: i.e. past tense when reporting data and present tense for discussion and interpretation

Line 44: been reported

Line 68: typically “de-repress” is used

Response: Thanks for your good suggestion. According to your advice, we have revised the use of the articles and the use of tenses in the manuscript, changed “It is also reported” to “It has been reported”, and changed “depress” to “repress” in the revised manuscript. Additionally, we have revised the English of manuscript through English language editing service called Editage at www.editage.cn.

EDITOR COMMENTS:

As well as the reviewer comments, we ask that you cite and discuss the manuscript 'Quantitative Imaging of D-2-Hydroxyglutarate in Selected Histological Tissue Areas by a Novel Bioluminescence Technique' – <https://www.frontiersin.org/articles/10.3389/fonc.2016.00046/full>.

Response: Thanks for your good advice. According to your suggestion, we have cited and discussed the reference “Quantitative Imaging of D-2-Hydroxyglutarate in Selected Histological Tissue Areas by a Novel Bioluminescence Technique” in the revised manuscript as follows:

“A bioluminescence technique based on NAD-dependent D-2-HG dehydrogenase, NADH:FMN-oxidoreductase, and luciferase has also been developed for the *in situ* detection of D-2-HG⁴³. This technique allows microscopical determination of D-2-HG in sections of snap-frozen tissue with a detection range of 0–10 $\mu\text{mol g}^{-1}$ tissue (wet weight).”

In the section “References”:

43. Voelxen, N.F. *et al.* Quantitative imaging of D-2-hydroxyglutarate in selected histological tissue areas by a novel bioluminescence technique. *Front. Oncol.* **6**, 46 (2016).

Reviewers' Comments:

Reviewer #1:

Remarks to the Author:

The authors have satisfactorily addressed all the comments and issues raised in the previous round of review.

Reviewer #2:

Remarks to the Author:

The authors have addressed my previous comments.

Reviewer #3:

Remarks to the Author:

The authors have responded to my comments and concerns in a satisfactory manner and the suggested experiments have been conducted. This reviewer has only some minor comments that should be addressed.

Line 96: "however, the non-specificities of their possible effectors should not be overlooked". This sentence is unclear.

General comment for all numerical values provided. The number of decimal places of values should be realistic and in agreement with the error value. For example the authors report a V_{max} of $40,681.73 \pm 946.03$ microM/min. Considering the error associated, the value should be $40,681 \pm 946$. All values should be revised and corrected to realistic measurements.

Line 386: better "responsive regulator"

Line 581: "The compound which ΔT_m value > 2 °C is considered the ligand of DhDR" The sentence needs some rephrasing. It should be stated that compounds that increase the T_m by more than 2 degrees are considered hits, but definite proof of ligand binding has to be obtained by ITC. This is necessary to specify since there are examples of false positive measurements in thermal shift assays, i.e. T_m increases by more than 2 degrees that were not due to ligand binding but additional effects such as ligand induced pH changes causing increases in protein stability.

Supp. Fig. 18. Better "...of purified *P. aeruginosa* WbpB"

Supp. Fig. 20: There are no error bars associated with these values.

Supp. Table 3: Suggestion for Table legend: Assessment of the robustness of biosensor performance. Interference with biosensor function by a number of physiologically relevant compounds (at 100 microM).

Responses to the reviewers' comments point-by-point

NCOMMS-21-12376A

Thanks a lot for the reviewers' comments, which are very useful for us to improve our manuscript. With regard to reviewers' comments and suggestions, we reply as follows:

REVIEWER COMMENTS

Reviewer #1 (Remarks to the Author):

The authors have satisfactorily addressed all the comments and issues raised in the previous round of review.

Response: Thank you very much for your time on our paper.

Reviewer #2 (Remarks to the Author):

The authors have addressed my previous comments.

Response: Thank you very much for your time on our paper.

Reviewer #3 (Remarks to the Author):

The authors have responded to my comments and concerns in a satisfactory manner and the suggested experiments have been conducted. This reviewer has only some minor comments that should be addressed.

Response: Thanks a lot for your positive comments. We have made corresponding changes in the revised manuscript according to your kind suggestions.

Line 96: "however, the non-specificities of their possible effectors should not be overlooked". This sentence is unclear.

Response: Thanks for your reminding. We have rephrased this sentence in the

revised manuscript as follows:

“Thus, these two transcriptional regulators in *P. thermoglucosidasius* DSM 2542 and *B. cereus* NJ-W may respond to D-2-HG and regulate D2HGDH expression but lactate and glycolate may also be their effectors.”

General comment for all numerical values provided. The number of decimal places of values should be realistic and in agreement with the error value. For example the authors report a V_{max} of $40,681.73 \pm 946.03$ microM/min. Considering the error associated, the value should be $40,681 \pm 946$. All values should be revised and corrected to realistic measurements.

Response: Thanks for your good suggestion. We have checked the numerical values provided in our manuscript and corrected them in the revised manuscript.

Line 386: better “responsive regulator”

Response: Thanks for your good suggestion. We have changed “specific regulator” to “responsive regulator” in the revised manuscript.

Line 581: “The compound which ΔT_m value > 2 °C is considered the ligand of DhdR”
The sentence needs some rephrasing. It should be stated that compounds that increase the T_m by more than 2 degrees are considered hits, but definite proof of ligand binding has to be obtained by ITC. This is necessary to specify since there are examples of false positive measurements in thermal shift assays, i.e. T_m increases by more than 2 degrees that were not due to ligand binding but additional effects such as ligand induced pH changes causing increases in protein stability.

Response: Thanks for your good suggestion. We have rephrased the sentence in the revised manuscript as follows:

“The compounds which ΔT_m value > 2 °C are considered hits and investigated subsequently by isothermal titration calorimetry (ITC) to obtain the definite proof of ligand binding.”

Supp. Fig. 18. Better “...of purified *P. aeruginosa* WbpB”

Response: Thanks for your good advice. We have revised the legend of **Supplementary Fig. 18** as follows:

“**Supplementary Figure 18. SDS-PAGE analysis of purified *P. aeruginosa* WbpB.** Lane M, molecular weight markers; lane 1, crude extract of *E. coli* BL21(DE3) harboring pETDuet-*wbpB*; lane 2, purified WbpB using a HisTrap column.”

Supp. Fig. 20: There are no error bars associated with these values.

Response: Thanks for your kind suggestion. The data presented in **Supplementary Fig. 20** is the gray value analysis of **Fig. 6e**, which is one representative experiment of three independent experiments. We have revised the legend of **Supplementary Fig. 20** in the revised Supplementary Information as follows:

“**Supplementary Figure 20. Relative quantification of O-antigen polymers in LPS of *P. aeruginosa* PAO1 and its derivatives.** The normalized gray values were obtained from the one representative experiment (**Fig. 6e**) of three independent experiments of silver-stained SDS-PAGE gel. The gray value analysis was performed by ImageJ 1.52p. Source data are provided as a Source Data file.”

Supp. Table 3: Suggestion for Table legend: Assessment of the robustness of biosensor performance. Interference with biosensor function by a number of physiologically relevant compounds (at 100 microM).

Response: Thanks a lot for your good advice. We have revised the table legend of **Supplementary Table 3** according to your suggestion.